# The crucial role of adhesion in the transmigration of active droplets through interstitial orifices

A. Tiribocchi [1] ✉, M. Durve [2], M. Lauricella [1], A. Montessori[3], D. Marenduzzo [4] & S. Succi[1,2,5]

Active fluid droplets are a class of soft materials exhibiting autonomous motion sustained by an energy supply. Such systems have been shown to capture motility regimes typical of biological cells and are ideal candidates as building-block for the fabrication of soft biomimetic materials of interest in pharmacology, tissue engineering and lab on chip devices. While their behavior is well established in unconstrained environments, much less is known about their dynamics under strong confinement. Here, we numerically study the physics of a droplet of active polar fluid migrating within a microchannel hosting a constriction with adhesive properties, and report evidence of a striking variety of dynamic regimes and morphological features, whose properties crucially depend upon droplet speed and elasticity, degree of confinement within the constriction and adhesiveness to the pore. Our results suggest that non-uniform adhesion forces are instrumental in enabling the crossing through narrow orifices, in contrast to larger gaps where a careful balance between speed and elasticity is sufficient to guarantee the transition. These observations may be useful for improving the design of artificial microswimmers, of interest in material science and pharmaceutics, and potentially for cell sorting in microfluidic devices.

Over the last decades, much research has been addressed on the active matter, an area of physics concerning systems whose internal constituents are capable of converting energy, adsorbed from the surrounding environment, into work or systematic movement[1,2]. A particular class of such systems is represented by active gels, densely packed soft materials in which the internal constituents have the tendency to assemble and align, producing structures with polar or nematic order[3–7]. Examples abound in biology, ranging from bacterial colonies[8–10] to actin filaments and microtubule bundles powered by motor proteins[11–14]. Such materials can be further divided into two broad classes, depending upon the structure of the fluid flow generated in their surroundings. In contractile materials, the fluid is pulled

inward axially and emitted equatorially, while in extensile ones the opposite holds[1]. Their inherent non-equilibrium nature fosters a wealth of sought-after phenomena with spectacular mesoscale collective behaviors, including spontaneous flows[12,15,16], turbulent-like motion in fluids with low Reynolds numbers[17], and unexpected rheological properties[18,19], to name but a few.

Of particular relevance to us are active fluid droplets, bio-inspired self-propelled emulsions whose autonomous motion is driven by a hierarchically-assembled active gel located within (either uniformly[20–22] or confined in a shell[23,24]) or adsorbed onto the interface[12]. Although distant from living cells, these droplets have been shown capable of capturing a number of features typical of cell

[1]Istituto per le Applicazioni del Calcolo CNR, via dei Taurini 19, 00185 Rome, Italy. [2]Center for Life Nano Science@La Sapienza, Istituto Italiano di Tecnologia, 00161 Rome, Italy. [3]Department of Engineering, Roma Tre University, Via Vito Volterra 62, 00146 Rome, Italy. [4]Scottish Universities Physics Alliance, School of Physics and Astronomy, University of Edinburgh, Edinburgh EH9 3JZ, UK. [5]Department of Physics, Harvard University, Cambridge, MA 02138, USA. ✉e-mail: a.tiribocchi@iac.cnr.it

dynamics, such as swimming[20], crawling[25], and spontaneous division[21], and have also served as a model tool for studying collective cell migration[26] and extrusion process in epithelial tissues[27]. Alongside their ability in describing the functioning of such complex biological processes, they could also offer a powerful platform for the design of programmable biomimetic soft materials[12,28] with enhanced mechanical properties, such as a higher control over the direction of motion and persistent motility, if compared to their passive analogs. These materials may be useful in a number of technological applications ranging from a pharmaceutics, as microscopic cargoes for the transport and release of drugs toward a diseased tissue[29], and food science, as carriers for the targeted delivery of the nutrients encapsulated within[30], to material science for the design of engineered tissues[27].

In such circumstances, as well as under many physiologically relevant conditions (such as within capillary vessels[31]), a droplet often migrates through pore-sized constrictions, whose diameter is typically much narrower than that of the drop itself (of the order of tens of micrometers). From a fluid dynamics perspective, the crossing through interstices poses additional challenges with respect to a motion occurring in an unconstrained environment. In a purely passive system, the scenario is well established[32,33]. A Newtonian fluid droplet placed in an external flow would undergo deformations governed by the interplay between hydrodynamic interactions, favoring shape changes, and capillary forces, which oppose morphological modifications and tend to hold a spherical geometry. Their balance is controlled by the capillary number Ca = $\eta v/\sigma$, where $v$ is the droplet speed, $\sigma$ its surface tension $\eta$ the shear viscosity. An intense driving flow, for example, may favor the crossing through the pore if properly counterbalanced by a sufficiently high surface tension, which would guarantee the stability of the drop and avoid its rupture.

The inclusion of an active gel could substantially change this picture. A non-homogeneous distribution of such material onto the fluid interface, for example, besides fostering the formation of coherent spontaneous flows[12], may concurrently alter the surface tension of the drop, thus considerably impacting morphology and mechanics when moving in a confined environment. If the active gel is encapsulated within, the drop is expected to harden, thus further opposing deformations. In addition, assembly and arrangement of the active material could decisively affect the elasticity of the drop and the structure of the fluid flow[20,25], hence imposing further constraints on the ability to migrate across a constriction.

Understanding their dynamics as well as their fluid–structure interactions, especially under controlled experimental conditions mimicking realistic environments, is thus essential for the optimal manufacturing of these materials. To make progress in this direction, the building of reliable computational models is often mandatory due to the complicated structure of the equations governing their physics[1,4,34]. Well-established numerical approaches (such as phase-field methods[35–37] and lattice Boltzmann algorithms[20,25,38,39]) combined with continuum theories have provided robust machinery to model the dynamics of active gel droplets in pore-free geometries[20,26,40–43]. However, their motion through narrow interstices has not been sufficiently investigated so far.

In this paper, we numerically study the transmigration of a droplet of active polar fluid across a constriction, following a design inspired by a typical lab-on-chip microfluidic device[44–46]. This one is modeled as a long thin channel hosting a pore-like interstice made of two solid pillars glued to opposite flat walls. The theoretical framework to investigate the physics of the transmigration is based on a phase-field-like approach, whose details are illustrated in the section Methods. It basically consists of a set of phase fields $\phi_i(\mathbf{r}, t)$ ($i = 1, 2, 3$), where $\phi_1(\mathbf{r}, t)$ accounts for the density of the active material encapsulated within our active droplet, while $\phi_2(\mathbf{r})$ and $\phi_3(\mathbf{r})$ are two *static* fluid-free fields modeling the pillars of the constriction[42]. The active material is a contractile gel whose mesoscopic orientational order is captured by a

liquid crystal vector field $\mathbf{P}(\mathbf{r}, t)$, while the global fluid velocity is represented by a further vector field $\mathbf{v}(\mathbf{r}, t)$. The dynamics of $\phi_1$ and $\mathbf{P}(\mathbf{r}, t)$ are governed by advection-relaxation equations while that of the fluid velocity $\mathbf{v}(\mathbf{r}, t)$ obeys the Navier–Stokes equation[47]. The equilibrium properties of this system are described by a Landau–de Gennes free-energy functional[48], augmented with a repulsive term between droplet and pillars plus a contribution favoring adhesion between the fluid interface and the pore.

Extensive lattice Boltzmann simulations show a rich variety of dynamic regimes whose physics is controlled by (i) the ratio between the size of the constriction and droplet diameter, (ii) speed and elasticity (including interfacial tension and polar field deformations) of the droplet, and (iii) adhesiveness between pillars of the pore and droplet itself. Central to these results is that such adhesion forces are decisive to enable the transmigration, especially for narrow interstices. Indeed our findings support the view that, while for wide pores the crossing is guaranteed by a careful balance between droplet speed and elasticity, for smaller ones it is generally forbidden unless adhesion forces come into play, provided that at the pore entry, they are higher than at the exit. Within the orifice, the droplet is found to display a series of shape deformations (from ampule-like to hourglass geometries) whose stability is controlled by the interplay between fluid velocity, exhibiting short-lived rectilinear flow, and elasticity of fluid interface and contractile gel, hosting splay and bend liquid crystal distortions. The minimal design of our computational model might suggest that the functioning of biomimetic droplet-based materials could rely exclusively upon mesoscale physics-based machinery rather than on complex microscopic multi-body interactions governing the physics at lower length scales.

## Results
### Motile droplet within a microfludic channel
The mechanism leading to the self-propulsion of a droplet comprising a contractile material, such as a network of actin filaments cross-linked with myosin proteins, has been theoretically investigated in previous works[20,21,25,35,36,40,43,49]. Following refs. [20,25], we consider a 2D mixture in which the active material is described in terms of a liquid crystal whose mean orientation is captured by a polar field while the contractile effect is modeled, at a mesoscale level, via a stress term (see the section "Methods" for further details). This one takes the form $\sigma_{\alpha\beta}^{\text{active}} \sim -\zeta \phi_1 P_\alpha P_\beta$, which is invariant under global polarity inversion and whose strength is gauged by the activity $\zeta$, negative for contractile mixtures. If $\zeta$ exceeds a threshold value, the active stress causes a spontaneous flow which breaks the inversion symmetry and sets the droplet into motion, along a direction controlled by an emerging splay deformation.

The essential steps of such dynamics are shown in Fig. 1. The droplet is placed within a microfluidic channel and is initialized as a circular region where $\phi_1 = \phi_0$ inside and $\phi_1 = 0$ outside (Fig. 1a). The polar field is initially uniform and aligned along the $y$ direction within the droplet (no anchoring of the polarization is set at the droplet interface), while it is zero outside. This means that the contractile gel is confined within the active drop and is polarized, while the surrounding region represents an isotropic Newtonian fluid. The activity $\zeta$ is then turned on and is set at a value allowing for the motion of the droplet. Before attaining a motile state, the drop temporarily elongates perpendicularly to the polarization $\mathbf{P}$ remaining motionless, an effect caused by the competition between interfacial tension, opposing shape deformations, and contractile stress, favoring hydrodynamic instability (see Fig. 1b). In this condition, the fluid flow surrounding the droplet acquires a four-roll structure (it is pulled inward equatorially and emitted axially, see Fig. 1f), thus preventing any net motion. Such an arrangement essentially results from the sum of the dipolar hydrodynamic flows formed around each contractile unit (such as the actomyosin complex, Fig. 1d). Indeed, at the microscopic level, the

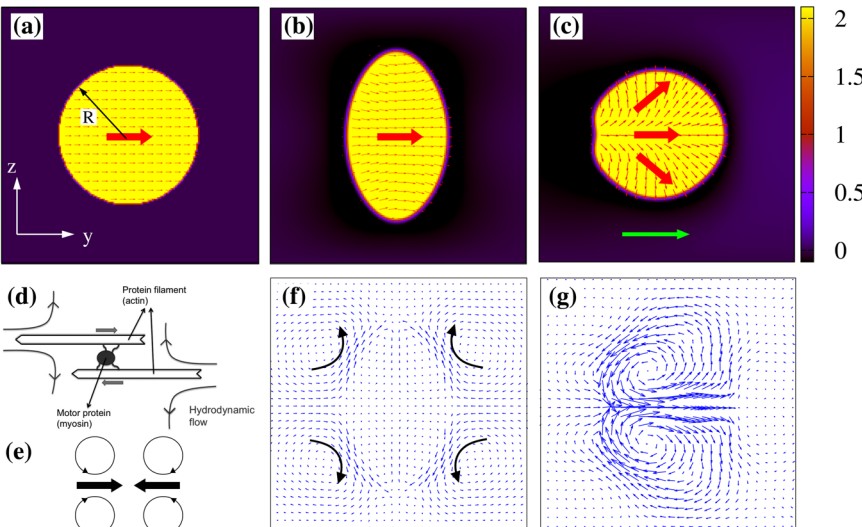

**Fig. 1 | Shapes and velocity field of a motile contractile droplet. a** Initial configuration of an active droplet. The red arrows indicate the direction of the polarization field **P**. The droplet is placed within a microfluidic channel of size $L_y = 500$ and $L_z = 170$. Here only a portion of the lattice is shown. **b** Intermediate pre-motile state of the contractile suspension with $\zeta = -8 \times 10^{-4}$. The droplet elongates perpendicularly to the direction of the polarization which remains essentially parallel to the $y$ direction. **c** However, this value of $\zeta$ is sufficiently high to destabilize the polarization, which gives rise to a large splay deformation. Once this occurs, the drop acquires a unidirectional motion along the direction indicated by the green arrow. **d** Schematic view of the hydrodynamic flow produced by a contractile material, such as actomyosin. The myosin protein pulls two actin filaments along opposite directions (indicated by tick gray arrows), yielding a four-roll flow in their surroundings. Panel (**d**) is adapted from[20]. **e** A minimal model of the force dipole produced by a contractile material. The thick black arrows indicate the direction of the force dipole while the circles represent the emerging four vortices of fluid. **f** and **g** Velocity field of the pre-motile (**b**) and motile (**c**) states. In the former, the fluid is pulled inward along the equator (parallel to the direction of **P**) and emitted axially (perpendicularly to **P**), giving rise to a macroscopic four-vortex structure. In the latter, a splay distortion fosters the formation of two counter-rotating vortices pushing the drop forward. The droplet radius at equilibrium is $R = 45$ lattice sites and the color map represents the value of the order parameter $\phi_1$, ranging between 0 (black) and 2 (yellow).

motor protein would pull two protein filaments together (Fig. 1e), causing an inward force pair that produces a contractile stress (see also the section "Methods"). This process also modifies the direction of the polarization, which remains basically uniform in the bulk but slightly deforms near the droplet interface, where a preferential perpendicular orientation emerges almost everywhere except at the ends of the elongated drop. The active anchoring, genuinely induced by the contractile stress[50], will persist in the following motile state, although its orientation will be considerably affected by the confinement conditions (especially when the drop migrates through narrow interstices, see the next sections).

Afterward, the non-motile configuration becomes unstable with respect to splay distortions, since the contractile stress is high enough to overcome the resistance to deformation mediated by the elastic constant $\kappa$. A suitable dimensionless quantity controlling the balance between activity and elasticity is the Ericksen number $\mathrm{Er} = \zeta R^2/\kappa$ ($R$ is the droplet radius) which ranges approximately from 5 to 50 in our simulations (see Supplementary Notes 1 and 2 for details on parameter values), thus high enough to destabilize the droplet and lead to its motion. The vectors of **P** then fan outwards arranging into a typical liquid crystal splay deformation (where $\nabla \cdot \mathbf{P} > 0$), while the droplet starts to move along the direction set by **P** (see Fig. 1c) sustained by two symmetric counter-rotating vortices (see Fig. 1g)[20]. This motion would last over long periods of time and would proceed unidirectionally with a steady velocity, typical of an active droplet swimming in a Newtonian fluid in the absence of external perturbations or constraints. We note that such spontaneous motion has been found to partially model the dynamics of tumor cells moving inside an elastic gel[20,51], where self-motility is solely triggered by myosin contractility rather than other mechanisms such as actin polymerization, usually essential in crawling cells[52].

The picture described so far dramatically changes when a droplet migrates in a complex environment flowing, for example, across a constriction of size much narrower than that of the droplet itself. In the next section, we precisely investigate this process providing an accurate description of the fluid-structure interaction along with a minimal set of key physical ingredients controlling the transmigration.

## Motile droplet across a wide constriction

We start off by considering the migration across a constriction of size $h$ comparable with the diameter $D$ of the active droplet. In Fig. 2a–f (and Supplementary Movie 1) we show a time sequence of such a process for $\lambda = h/D \simeq 0.8$, where $\lambda$ is the confinement parameter. The droplet is initialized as in the pore-free case, i.e. a circular region where the polarization is initially parallel to the $y$-direction. Once the activity is turned on ($\zeta = -8 \times 10^{-4}$), the droplet elongates in the direction perpendicular to **P** and then acquires a unidirectional motion at constant velocity (Figs. 2a and 3a, d, where the position and speed of the center of mass are plotted), a process akin to that described in the previous paragraph. In the vicinity of the pore (modeled placing two solid pillars at distance $h$, see the section "Methods"), it slightly squeezes and stretches forward (Fig. 2b), while its speed gradually diminishes up to a minimum value, attained approximately once the leading edge enters the gap (Fig. 3g, h). However, this slowdown does not arrest the motion, which proceeds favored by a series of weak morphological deformations sufficient to boost the droplet and guarantee the transmigration. Within the orifice, the droplet undergoes a light longitudinal stretching and compression (Fig. 2c) fostering an increase of speed of approximately three times higher than the value at the entrance of the pore (Fig. 3i), followed by a mild decompression (Fig. 2d) where the speed goes back to its steady unconstrained value (Fig. 3j). Afterwards, the droplet expands (Fig. 2e) restoring the typical crescent-like shape (Fig. 2f) observed out of the pore. During the process, the polarization remains basically unaltered, preserving its splay arrangement kept for the entire course of the migration.

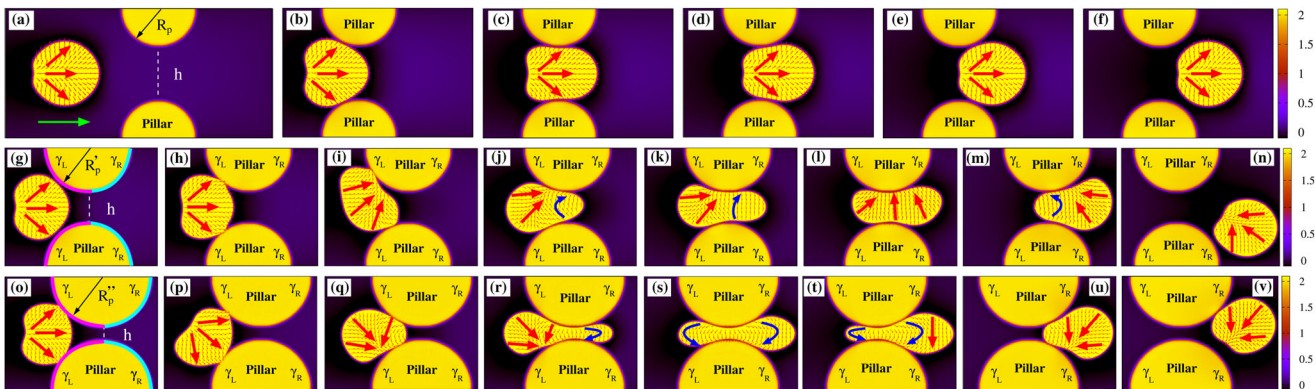

**Fig. 2 | Transmigration of an active drop across a constriction. a–f** If the size of the constriction is comparable with that of the droplet ($\lambda \simeq 0.8$, $h = 72$, $R_p = 49$, $D = 90$), the latter moves unidirectionally (the green arrow indicates the direction of motion) undergoing weak shape deformations, such as a slight longitudinal stretching (**b**) and mild compression (**c** and **d**). Out of the constriction, the circular shape is restored (**e** and **f**). The splay distortion (highlighted by large red arrows) of the polarization remains essentially unaltered. **g–n** If $\lambda \simeq 0.5$ ($h = 46$, $R'_p = 62$, $D = 90$), the incipient unidirectional motion slows down as the droplet approaches the constriction, where large portions of the fluid interface stick to (**g** and **h**) because of adhesion forces, larger at the entry and weaker at the exit (here $\gamma_L = 3 \times 10^{-2}$, $\gamma_R = 7.5 \times 10^{-3}$). Afterwards, the droplet undergoes an intense folding (**i**) followed by a significant elongation (**j–l**) and decompression (**m**). Within the constriction, the polarization aligns essentially perpendicularly to the direction of motion, exhibiting a temporary bend deformation (highlighted with blue arrows), initially at the front (**j** and **k**) and then shifted towards the back (**m**). At the exit of the pore, the droplet detaches from the pillars and proceeds along the direction imposed by the splay deformation (**n**). **o** and **v** If the size of the constriction is very narrow ($\lambda \simeq 0.2$, $h = 20$, $R''_p = 75$, $D = 90$), the droplet initially adheres to the pillars and then shifts downwards to protrude its leading edge within the pore (**o–q**). Afterwards, it dramatically stretches along the direction of motion acquiring an initial ample-like shape (**r**), subsequently replaced by an intermediate hourglass structure (**s**). Finally, the droplet retracts its rear, pushes its front forward (**t**), and leaves the pore (**u** and **v**), a process facilitated by the lower adhesion forces at the exit (here $\gamma_L = 2.5 \times 10^{-2}$, $\gamma_R = 10^{-2}$). Within the pore, the polar field shows long-lasting bend distortions, either along with splay deformations (**r–t**) or alone and spread to the whole drop (**s**).

## Motile droplet across a medium size constriction

Decreasing $\lambda$ enough can permanently hinder the crossing through the pore. This is shown in Supplementary Movie 2 where we simulate the dynamics of an active droplet swimming within a microchannel with $\lambda \simeq 0.5$. The droplet initially self-propels forward following the mechanism previously described and, once near the pore, hits the pillars, which halt its motions impeding the transmigration. Note that, despite the stop, the droplet partially preserves a crescent-like shape, due to the permanent splay distortion of the liquid crystal caused by the contractile activity.

These results do not automatically rule out the possibility to observe a crossing, an event that may occur, for example, if the activity $|\zeta|$ is higher and the surface tension is lower than the values considered so far. The former would increase the droplet speed and ensure a stronger impact against the pore, thus likely providing the necessary force to squeeze in, while the latter would diminish the resistance to undergo considerable shape deformations favoring the crossing. In Supplementary Movie 3, we show, for example, the dynamic behavior of a contractile droplet where $\zeta = -10^{-3}$ and $\lambda \simeq 0.6$. Despite raising $|\zeta|$, the force is not high enough to guarantee the crossing. Indeed, once near the pore, the droplet moves upwards, stretches longitudinally, and turns back along the direction imposed by the splay deformation, only temporarily lost during the previous elongation in which **P** becomes approximately uniform. Further increasing $|\zeta|$ would require the adjustment of other thermodynamic parameters to ensure a correct balance between splay distortions (controlled by $\kappa$, see Eq.(1)) and interfacial tension (controlled by $a$ and $k$) in order to observe a unidirectional motion.

An alternative route potentially favoring the transmigration is through adhesion forces enhancing the connectivity between the active droplet and the pillars of the constriction. Following microfluidic experiments on the transmigration of real cells[44,46], an adhesive effect could be promoted by functionalizing the pillars with various proteins, such as fibronectin or collagen, while the surrounding area (i.e. the flat walls) would be passivated using chemical repellents. This mechanism could (i) minimize the bounce back of the droplet, (ii) provide the additional interfacial stress necessary to move the droplet

forward as in contact with solid surfaces and iii) facilitate substantial morphological deformations under strong confinement. Such a strategy draws partial inspiration from that of eukaryotic cells crawling on a solid substrate[53], a process in which the anchoring of the actin cytoskeleton to the surface is controlled by the focal adhesions, clusters of membrane proteins continuously assembled at the cell front and disassembled at the rear during the gliding[52]. Although our active droplet remains distant from a living cell in many aspects, nonetheless it may provide a model for lamellar cell fragments[54,55] (deprived of the nucleus) or for biomimetic artificial cells with a propensity to self-propel[12] and capable of crossing micropores with chemically functionalized adhesive surfaces[56].

In Fig. 2g–n (and Supplementary Movie 4) we show the dynamics of a motile droplet ($\zeta = -7 \times 10^{-4}$) crossing a constriction where $\lambda \simeq 0.5$ and in the presence of adhesive forces between the interface of the drop and the surface of the pillars. In our model the strength of the adhesion is controlled by the positive constant $\gamma$ (see Eq.(1)), which we set equal between the drop and each pillar and patterned following the sketch reported in Fig. 2g. We essentially define two values of $\gamma$, namely $\gamma_L$ and $\gamma_R$ gauging the adhesion of the left and right sides of the pore, with the general constraint that $\gamma_L > \gamma_R$ and such that $\gamma = \gamma_L$ for $0 < y \leq l/2$ and $\gamma = \gamma_R$ for $l/2 < y < l$, being $y$ the horizontal coordinate and $l$ the diameter of the pillars. In addition $\gamma_{\min,L} \leq \gamma_L \leq \gamma_{\max,L}$ and $\gamma_{\min,R} \leq \gamma_R \leq \gamma_{\max,R}$, where $\gamma_{\min,L}$, $\gamma_{\max,L}$, $\gamma_{\min,R}$, $\gamma_{\max,R}$ represent critical values depending on the details of the simulations (such as speed of the drop, elasticity, and size of the pore) beyond which the crossing is generally inhibited. In Fig. 2g–n we have $\gamma_L = 3 \times 10^{-2}$ (with $\gamma_{\min,L} \simeq 2 \times 10^{-2}$, $\gamma_{\max,L} \simeq 5 \times 10^{-2}$) and $\gamma_R = 7.5 \times 10^{-3}$ (with $\gamma_{\min,R} \simeq 5 \times 10^{-3}$, $\gamma_{\max,R} \simeq 2 \times 10^{-2}$, see also Supplementary Note 3 for further results). Such a design essentially allows for higher adhesion forces at the entrance of the constriction and weaker ones at the exit, thus potentially enabling transmigration. In Supplementary Note 4 we show that the physics remains qualitatively similar if a smoother variation of $\gamma$ between the entry and exit of the constriction is considered.

Once the droplet approaches the pore, small portions of its interface hit opposite pillars and adhere to their surfaces (Fig. 2g, h), thus causing a progressive slowdown and a light deviation from the

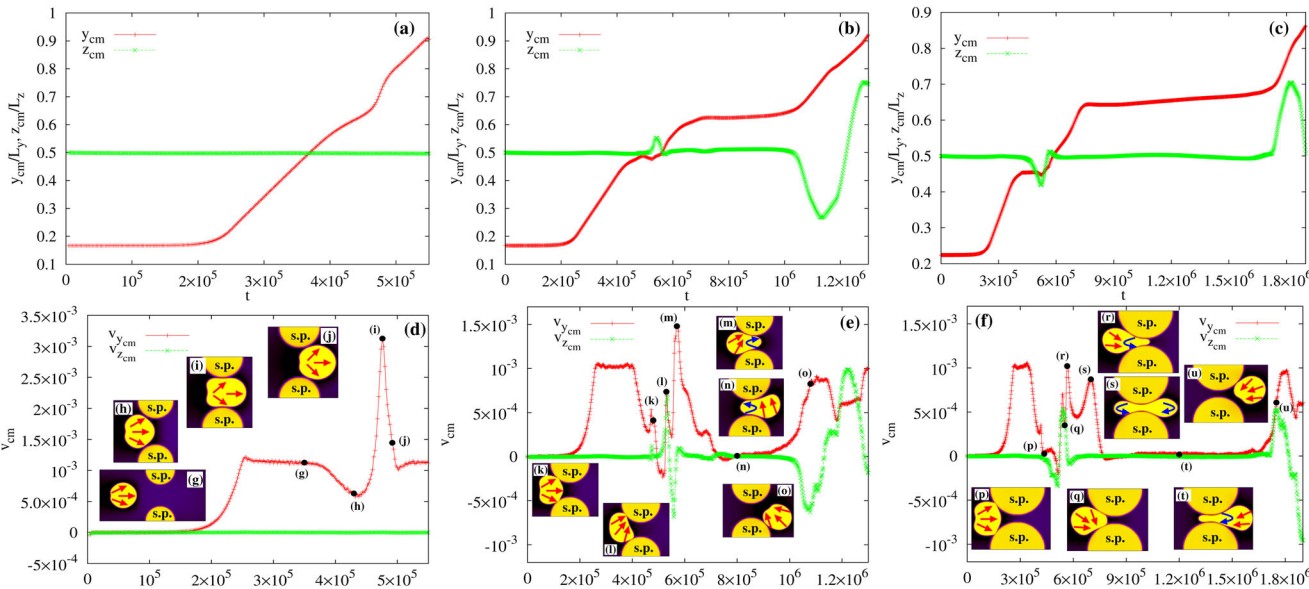

**Fig. 3 | Center of mass position and speed of the active drop for different pore sizes.** Top row: Time evolution of the $y$ (red, pluses) and $z$ (green, crosses) components of the center of mass of the active droplet for $\lambda \simeq 0.8$ (**a**), $\lambda \simeq 0.5$ (**b**), and $\lambda \simeq 0.2$ (**c**). Bottom row: Time evolution of $y$ (red, pluses) and $z$ (green, crosses) components of the speed of the center of mass for $\lambda \simeq 0.8$ (**d**), $\lambda \simeq 0.5$ (**e**), and $\lambda \simeq 0.2$ (**f**). Black dots indicate the position of the insets representing instantaneous configurations observed during the crossing. Red arrows show the direction of the splay deformations while blue ones that of the bending. Also, "s.p." stands for solid pillars. If $\lambda \simeq 0.8$, the droplet proceeds almost unaltered along its unidirectional trajectory, progressively decreasing the speed at the entrance of the pore (**g**, **h**) and rapidly augmenting it in the middle (**i**, **j**), where shape deformations are larger. If $\lambda \simeq 0.5$, the speed diminishes at the entrance of the constriction (**k**) and then rapidly

augments (**l**, **m**), as the droplet front squeezes into the pore. Afterward, the speed undergoes a second sharp decrease (**n**), considerably slowing down the droplet but only temporarily arresting its motion. It gradually starts over (**o**) due to internal fluid flows caused by the contractile material. Note that, alongside the usual splay deformation, temporary bend distortions emerge (blue arrows), initially located at the front and progressively shifted backward. If $\lambda \simeq 0.2$, once again the decrease of the droplet speed at the entrance of the pore (**p**) is followed by its quick rise once the tip of the drop squeezes in (**q**, **r**) and moves forward (**s**). Here, bend deformations persist longer than before and spread over the entire drop in the middle of the pore. Then, the speed undergoes a second quick reduction freezing the drop into an ampule-like shape (**t**) for a long period of time, after which the transmigration is completed (**u**).

rectilinear trajectory (see Fig. 3b and k). Afterward, the drop moves upward due to an internal fluid vortex (see the section on the fluid–structure for a detailed description), an effect not sufficient to determine its detachment from the pillars but crucial to drive morphological changes necessary to squeeze into the gap (Fig. 2i–l). Indeed, the droplet initially stretches pushing its front within the constriction and then elongates longitudinally dragging the rear, thus causing an increase in its perimeter (see the next paragraph about the energetic balance). On the contrary, the droplet area (i.e. $\phi_1$) is conserved, since its evolution is governed by model B-like dynamics (see the section "Methods"). During such process, the speed undergoes a sharp increase (Fig. 3l, m) followed by a steep reduction (Fig. 3n), yielding a temporary freezing of the droplet shape into a peanutlike structure, where opposite sides of its interface remain firmly anchored at the surface of the pillars. However, adhesion forces at its front are weaker than those at the rear ($\gamma_R < \gamma_L$), an effect that can facilitate crossing and detachment of the droplet if a sufficient propulsion force operates. This is precisely the dynamics observed in the final stage of the process, where the droplet slowly decompresses while leaving the pore (Fig. 2m, n) and its center of mass speed raises once again (Fig. 3o). Finally note that, along with splay distortions (generally the dominant contribution far from the constriction), the confined environment of the pore triggers the formation of regions where the polar field exhibits temporary bend deformations (Fig. 3m, n, blue arrows), an arrangement generally emerging as an elastic instability in extensile material[l], here easier to accommodate in such a highly stretched geometry. Interestingly, even though the interface anchoring remains largely perpendicular, a tangential orientation arises where droplet elongation increases, an effect mainly observed at the entry and exit of the pore.

The results discussed so far suggest that higher values of adhesion force at the entrance and lower ones at the exit of the gap can favor transmigration. If alternatively, the adhesion between the droplet interface and the surface of the pillars is uniform everywhere the crossing can be inhibited, as shown in Supplementary Movie 5 where $\lambda \simeq 0.5$, $\zeta = -8 \times 10^{-4}$, and $\gamma_L = \gamma_R = 0.03$. Once again, a series of shape modifications, driven by a combination of contractility, elastic deformations of polarization, and fluid interface plus adhesion forces, allows the active droplet to squeeze into the pore. However, higher values of $\gamma_R$ at the exit of the constriction prevent the migration, permanently sequestering the droplet in the middle of the gap. Note that this outcome is in agreement with the constraint on $\gamma_R$, since here $\gamma_R > \gamma_{\max,R}$.

## Motile droplet across a narrow interstice

Further diminishing $\lambda$ leads to a higher complex behavior where adhesion forces, once more, are found to play a decisive role. In Fig. 2o–v (and Supplementary Movie 6) we show the time evolution of an active droplet crossing a constriction with $\lambda \simeq 0.2$, $\zeta = -7 \times 10^{-4}$, $\gamma_L = 2.5 \times 10^{-2}$ and $\gamma_R = 10^{-2}$. Here, $\gamma_{\min,L} \simeq 2 \times 10^{-2}$ and $\gamma_{\max,L} \simeq 4 \times 10^{-2}$, while $\gamma_{\min,R} \simeq 5 \times 10^{-3}$ and $\gamma_{\max,R} \simeq 1.5 \times 10^{-2}$. Note that the narrowing of the pore shrinks the range of values of $\gamma$ enabling the crossing. The initial stage of the process follows dynamics akin to that observed for a mild constriction ($\lambda \simeq 0.5$). Once portions of interfaces adhere to the surface of the pillars (Fig. 2o), the droplet slows down (Fig. 3p) and shifts downwards (Fig. 2p) essentially preserving the arrangement of the internal polarization. Afterward, the speed increases (Fig. 3q–s) and the front squeezes into the interstice (Fig. 2q) but, unlike the pore of larger size, here the droplet undergoes a dramatic stretching. Indeed, it

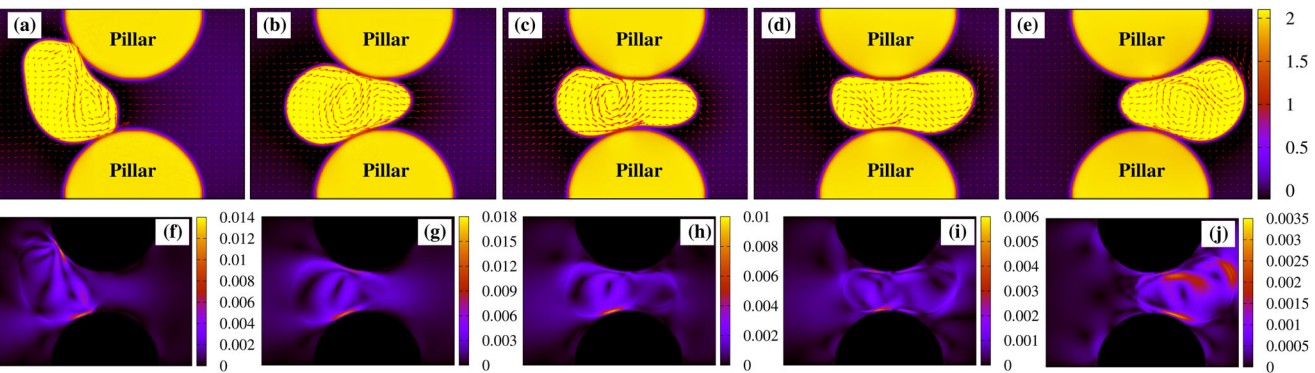

**Fig. 4 | Velocity field in a medium size constriction.** The top row **a**–**e** shows the structure of the velocity field **v** within and in the surrounding of an active droplet crossing a constriction where $\lambda \simeq 0.5$, while the bottom one shows its magnitude |**v**|. The double vortex pattern observed in the unconstrained motile droplet (see Fig. 1g) turns into a single one rotating counterclockwise (**a**), with magnitude larger near the pillars (**f**). Once the droplet enters the pore, such vortex shifts towards the center of the drop, while a net unidirectional flow emerges at the front (**b**, **c**) and becomes dominant near the exit (**d**), where it acquires an oscillating structure. During the crossing, the magnitude |**v**| remains higher near the pillars (**g**–**i**), whereas it considerably decreases at the exit (**j**), once the double vortex structure is restored (**e**).

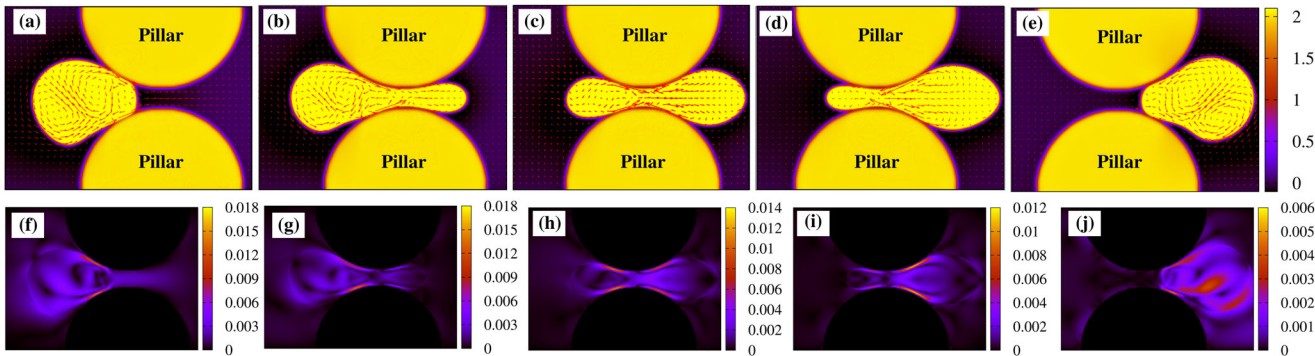

**Fig. 5 | Velocity field in a narrow constriction.** The top row **a**–**e** shows the velocity field **v** during the transmigration with $\lambda \simeq 0.2$, while the bottom one shows its magnitude |**v**|. Here, the double vortex pattern (**a**) moves progressively backward (**b**) until it is temporarily replaced, in the middle of the pore, by a squeezed four-fold symmetric structure, in which a unidirectional flow pushing the drop forward still prevails (**c**). Once the droplet leaves the pore, the double vortex gradually recovers, initially confined at the front (**d**) and afterward spread within the bulk (**e**). Once again, the magnitude of **v** remains high especially near the surfaces of the pillars (**f**–**i**), while substantially diminishing as the droplet leaves the constriction (**j**).

provisionally acquires an ampule-like shape (Fig. 2r) subsequently replaced by an hourglass structure made of two rounded blobs of approximately similar size located near the entrance and the exit of the pore (Fig. 2s). Then, its rear retracts within the orifice while the front protrudes out of the pore and broadens, leading to a long-lasting ampule shape moving at very low speed. At this stage, the velocity sharply diminishes (Fig. 3t) basically because of a lack of sufficient propulsion fostered by large splay distortions. Note in particular that, during the course of the crossing, a relevant bend deformation emerges, initially solely at the front (Fig. 2r), then spread to the whole droplet (Fig. 2s), and finally confined at the back (Fig. 2t) together with splay distortions. At the fluid interface, especially nearby the two bulges, the polar field displays a preferential tangential orientation, an effect sharper than that observed in wider constrictions due to the narrowing of the gap. At the exit, the lower adhesion forces between the surface of the pillars and the interface allow the droplet to gain a high enough speed (Fig. 3u) and leave the constriction (Fig. 2u, v), while the shape progressively turns to circular and the splay distortion becomes dominant. It is worth highlighting that the speed reduction in the gap and the raise at the exit are generic features observed regardless of the size of the constriction, a result in agreement with experiments of transmigration of 3d cells[44].

## Fluid–structure interaction

A deeper understanding of the dynamics of the transmigration can be gained by the evaluation of the fluid–structure interaction, especially for mild and narrow constrictions where morphological deformations are considerably higher than those observed in larger pores. In Figs. 4 and 5 we show the fluid velocity **v** (top row) and its magnitude |**v**| (bottom row) for $\lambda \simeq 0.5$ and $\lambda \simeq 0.2$, respectively, within the droplet and in the surrounding environment. Clearly, the structure of the fluid flow in these cases considerably departs from that of a droplet swimming in an unconstrained system (see Fig. 1g), where a couple of counter-rotating vortices sustain the motion. If, for example, $\lambda \simeq 0.5$, only a single counterclockwise vortex survives as the drop approaches the pore (Fig. 4a). During the transmigration, a unidirectional flow emerges at the front (Fig. 4b, c) pushing the vortex backward, until they merge producing a homogeneous oscillating pattern (Fig. 4d). Interestingly, a similar structure has been also observed in experiments of tumor cells in which the displacement within a micro-environment is driven by an osmotic pressure difference across the membrane, causing a net flow from the leading edge of the cell to the rear[57]. Note that the magnitude of the velocity is particularly high at the interfaces in contact with pillars (Fig. 4f–i), an effect indicating that the adhesion is crucial to provide the excess kinetic energy necessary to push the droplet within the constriction and enable its transmigration.

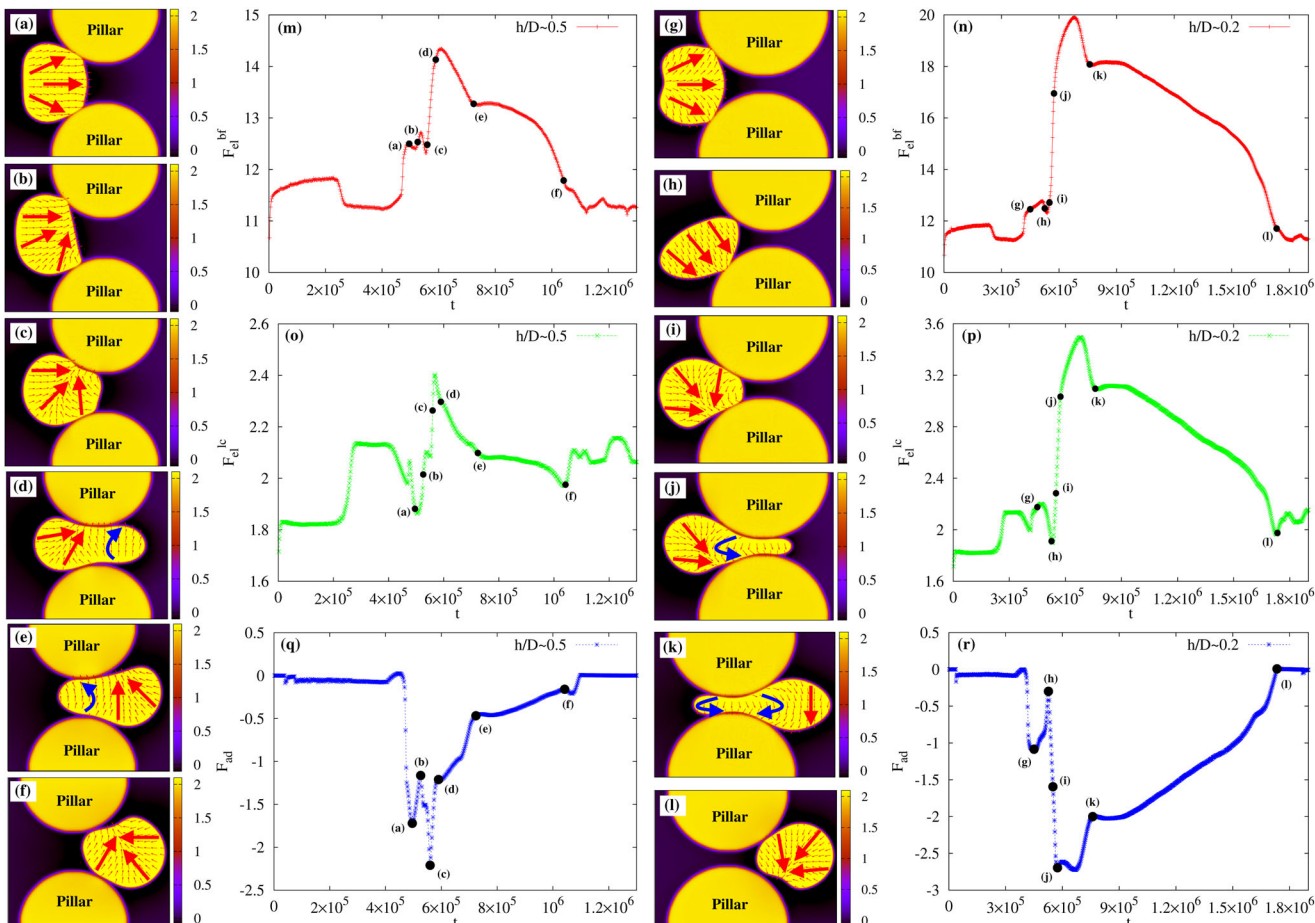

**Fig. 6 | Free energy.** The plots show the time evolution of the elastic free energy $F_{el}^{bf} = \int k/2(\nabla\phi_1)^2$ of the binary fluid (top line, red pluses), the elastic free energy $F_{el}^{lc} = \int \kappa/2(\nabla\mathbf{P})^2$ of the polar liquid crystal (green, crosses), and the adhesion contribution $F_{ad} = \int \sum_{i,j,i<j} \gamma \nabla\phi_i \nabla\phi_j$ (blue, asterisks). Black dots indicate the instantaneous configurations of the active droplet during the transmigration for $\lambda \simeq 0.5$ (left side) and $\lambda \simeq 0.2$ (right side). If $\lambda \simeq 0.5$, the droplet adheres to pillars and deforms (**a**–**c**), thus causing an increase of $F_{el}^{bf}$ (**m**) with respect to the values attained during the unconstrained motile state (for $t \leq 4.5 \times 10^5$). On the contrary, $F_{el}^{lc}$ (**o**) displays an initial descent due to a slight drop of splay followed by a quick growth, since bend distortions start to appear. Once the drop snakes into the pore (**d**), such contributions further augment since a higher stretching entails larger bend distortions, whereas near the exit they diminish (**e**, **f**), as the bend essentially disappears and the droplet reacquires a crescent-like shape with mild deformations. $F_{ad}$ (**q**) is zero when the drop is far from the pore, while it turns negative as its interface comes in contact with the pillars. In particular, $F_{ad}$ attains lower values during the squeezing and the subsequent crossing (**a**–**d**), whereas it gets progressively smaller once the drop approaches the exit and leaves the pore (**e**, **f**). If $\lambda \simeq 0.2$, $F_{el}^{bf}$ (**n**) out of the pore (**g**–**i**, **l**), $F_{el}^{lc}$ (**p**) and $F_{ad}$ (**r**) exhibit a behavior akin to that observed for a larger constriction while, during the crossing (**j**, **k**), their (absolute) values are generally higher, ultimately because shape deformations and liquid crystal distortions (including splay and bend contributions) are considerably heavier and persist over a longer period of time.

Once at the exit of the pore, the double vortex structure is restored (Fig. 4e) and $|\mathbf{v}|$ decreases about one order of magnitude within the drop (Fig. 4j).

If $\lambda \simeq 0.2$, once again the typical double vortex observed at the entrance of the gap shifts towards the rear as the drop sneaks into the pore (Fig. 5a, b). However, since the high confinement prevents the formation of fluid structures larger than the gap size, the vortices turn into a rectilinear flow exhibiting a four-fold symmetric structure (Fig. 5c). Such flow progressively weakens as the drop moves forward (a condition favored by the lower adhesion forces at the exit of the pore), and is gradually replaced by the double vortex, fully reestablished at the exit (Fig. 5d, e). As in the medium-size constriction, the highest values of the velocity field are found along the interfaces in contact with the pillars, where adhesion forces operate (Fig. 5f–j). Note finally that the velocity field, besides critically affecting the shape of the active droplet, profoundly depends on the structure of the underlying polar field, essentially regardless of the size of the constriction. In fact, while the double vortex pattern emerges in the presence of splay distortions, a rectilinear (often unidirectional) flow is produced when bend deformations are dominant.

The effects due to shape changes, modifications of the orientation of the active material as well as adhesion forces can be quantitatively gauged by computing the associated free energy contributions reported in Eq. (1). This is discussed in the next section.

## Energetic balance

In Fig. 6 we show the time evolution of $F_{el}^{bf} = \int k/2(\nabla\phi_1)^2$ assessing the interfacial energy of the droplet, $F_{el}^{lc} = \int \kappa/2(\nabla\mathbf{P})^2$ accounting for the deformations of the polarization and $F_{ad} = \int \sum_{i,j,i<j} \gamma \nabla\phi_i \nabla\phi_j$ quantifying the adhesion contribution. We start from $\lambda \simeq 0.5$ (left panel). As discussed in Fig. 1, once the activity $\zeta$ is turned on, the drop initially elongates axially attaining a motionless elliptical shape, and then acquires motion due to a spontaneous flow causing a symmetry breaking of the polar field. In the former regime both $F_{el}^{bf}$ (Fig. 6m) and $F_{el}^{lc}$ (Fig. 6o) increase and stabilize (for $t \leq 2 \times 10^5$), whereas in the latter (i.e. as the droplet starts to move) $F_{el}^{bf}$ lowers, since the drop turns to an approximately circular shape, and $F_{el}^{lc}$ augments due to the presence of splay deformations. Then they both attain a value kept constant until the drop approaches the pore (at $t \simeq 4.5 \times 10^5$). Note that in these regimes $F_{ad}$ (Fig. 6q) remains

essentially zero since drops and pillars are sufficiently far away from each other.

At the entry of the constriction, the droplet deforms (Fig. 6a–c) and elongates to squeeze in (Fig. 6d), thus causing a further increase of $F_{el}^{bf}$ (capturing the growth of interfacial area), which attains its maximum value approximately in the middle of the pore. On the contrary, $F_{el}^{lc}$ initially decreases since the splay distortion slightly weakens, and then it rapidly augments, especially when bend deformations appear at the front. Once at the exit of the pore, the droplet decompresses and the liquid crystal deformations turn milder (Fig. 6e, f), thus both free energy contributions gradually reduce to values comparable with those held before the crossing. During such a process $F_{ad}$ turns negative when the drop interface starts to adhere to the pillars, attaining its higher (absolute) values at the entry (Fig. 6a) and within the pore (Fig. 6c, d), basically because larger portions of interfaces are in close contact with the pillars.

If the size of the constriction decreases ($\lambda \simeq 0.2$, Fig. 6n, q, p), $F_{el}^{bf}$, $F_{el}^{lc}$ and $F_{ad}$ shows a time evolution akin to that discussed previously, albeit larger values are observed when the drop crosses the pore. Indeed, $F_{el}^{bf}$ and $F_{el}^{lc}$ rise, once again, at the entrance of the gap (Fig. 6g), slightly decrease later since elastic deformations get globally milder (Fig. 6h), and then considerably augment up to a maximum (almost doubling the values observed for a drop migrating freely in the microchannel), when the drop front squeezes in (Fig. 6i, j) and attains an hourglass shape. Afterward, $F_{el}^{bf}$ and $F_{el}^{lc}$ continuously diminish (Fig. 6k, l) until the drop has left the pore. As in the previous case, $F_{ad}$ turns negative once the interface adheres to the pillars, and its larger values are obtained basically when the drop displays the ampule and the hourglass shape.

## Dimensionless numbers

Further insights into transmigration can be gained by analyzing the physics in terms of a suitable set of dimensionless quantities. Common numbers used in droplet microfluidics are the Reynolds and capillary ones, defined as $Re = \rho v D/\eta$ and $Ca = v\eta/\sigma$, where $\rho$ is the fluid density, $v$ is the droplet speed, $\eta$ is the fluid viscosity and $\sigma$ is the surface tension (see also Supplementary Notes 1 and 2 for further details on specific values). The former represents the ratio of inertial forces to viscous ones and, in our simulations, is generally equal or below 0.1, thus well within the laminar regime. The latter measures the effect of viscous force (favoring shape deformations) versus surface tension ones (which oppose shape changes), and is approximately equal to 0.1 (or lower). This value ensures that a droplet breakup is an unlikely event.

In addition, we consider the following three quantities: the Ericksen number $Er = \zeta R^2/\kappa$, the adhesion number $A = \gamma_L/\gamma_R$, and the inertia over adhesion number $I_{A_{LR}} = \rho v^2 R^2/\gamma_{L,R}$. The former controls the dynamics at the onset of the spontaneous motion far from the constriction. More specifically, if $Er > 1$ the active forces are sufficiently large to overcome the elasticity of the liquid crystal (mediated elastic constant $\kappa$) and destabilize the droplet, finally inducing spontaneous motion. In our simulations, $Er$ is generally larger than 5, thus high enough to trigger self-locomotion. The adhesion number $A$ represents the balance between adhesion forces at the entry and the exit of the pore and, for successful transmigration, it must be strictly larger than 1 (since $\gamma_L > \gamma_R$). For $\lambda \simeq 0.5$, we get $2 \lesssim A \lesssim 10$, while for $\lambda \simeq 0.2$ we have $2 \lesssim A \lesssim 3$, a narrower range of values due to the reduction of the size of the pore. Finally, the number $I_{A_{LR}}$ gauges the importance of inertial forces over adhesive ones. For $\lambda \simeq 0.5$ and $\gamma_R = 5 \times 10^{-3}$ (a value for which the crossing occurs), one has $I_{A_R} \simeq 0.25$ and $0.025 \lesssim I_{A_L} \lesssim 0.06$ (assuming a droplet speed $v \simeq 5 \times 10^{-4}$). This means that, at the entry, inertial forces are much weaker than adhesive ones, a necessary condition to keep the droplet attached to the pillars and enable the crossing. The opposite is true at the exit, where lower adhesion forces

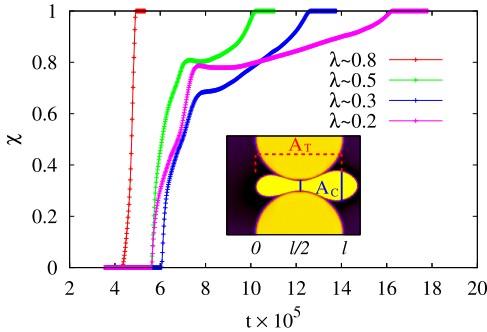

**Fig. 7 | Transmigration order parameter.** We plot the time evolution of $\chi(t) = A_C(t)/A_T(t)$, where $A_C$ represents the area fraction of droplet between the centerline (located at $l/2$) and the exit of the pore (placed at $l$), and $A_T$ is the total area of the droplet within the pore. If $\lambda \simeq 0.8$, $\chi(t)$ grows rapidly towards 1 following an approximately linear behavior. For decreasing values of $\chi$ one can distinguish three regimes: a fast-growing approximately linear one at the entry of the pore, a short stationary one (lasting longer for narrower interstices) with a temporary negative slope in the middle, and a final slow-growing one at the exit. The inner snapshot shows an instantaneous configuration of the transmigration with $\lambda \simeq 0.2$.

allow a droplet with sufficiently high speed to escape from the pore. For $\lambda \simeq 0.2$, similar considerations hold. Here, if $\gamma_R = 10^{-2}$, one has $I_{A_R} \simeq 0.1$ and $0.04 \lesssim I_{A_L} \lesssim 0.06$.

## Transmigration order parameter

Inspired by the Lubensky–Nelson model of polymer translocation through nanopores[58], before concluding we provide a characterization of the droplet transmigration in terms of a single order parameter-like quantity $\chi(t) = A_C(t)/A_T(t)$, defined as the ratio between the area fraction $A_C(t) = \int_{l/2}^{l} \phi_1(\mathbf{r},t)d\mathbf{r}$ of droplet that has transmigrated the centerline of a pore of length $l$ and the total area $A_T(t) = \int_{0}^{l} \phi_1(\mathbf{r},t)d\mathbf{r}$ of the droplet within the pore. It varies between 0 (the droplet has not passed the midline) and 1 (the whole droplet has overcome the midline). In Fig. 7 we show the time evolution of $\chi(t)$ for different values of $\lambda$, in simulations where $\zeta \simeq -7 \times 10^{-4}$, $2.5 \times 10^{-2} \leq \gamma_L \leq 3 \times 10^{-2}$ and $7.5 \times 10^{-3} \leq \gamma_R \leq 10^{-2}$. If the constriction is wide enough ($\lambda \simeq 0.8$), $\chi(t)$ exhibits a ballistic-like behavior rapidly growing towards 1 essentially with a single slope. On the contrary, if the size of the constriction diminishes, the crossing occurs over longer periods of time, which augments for decreasing values of $\lambda$. In these systems $\chi(t)$ displays basically three regimes, (i) a fast-growing approximately linear one at the entry of the pore, (ii) a transient stationary one roughly in the middle of the constriction, and (iii) a final slower monotonic regrowth at the exit. While in the first regime the transmigration proceeds rather quickly (more than half of the droplet has over overcome the midline, $\chi \geq 0.5$) basically because of a combination of droplet propulsion, splay deformations of the liquid crystal, and adhesion with the pore, later on, the process dramatically slows down and the droplet attains an almost non-motile state (see also Fig. 2). Here, the slope of $\chi(t)$ turns slightly negative (for $\lambda \simeq 0.5$ and $\lambda \simeq 0.2$) due to a temporary retraction of the droplet and then exhibits short-lived plateaus, lasting longer for smaller $\lambda$. In the last regime, the transmigration restarts and, as expected, occurs faster for larger $\lambda$ although at a speed much smaller than the one at the entry (in agreement with the results of Figs. 4 and 5).

Finally, computing $\chi(t)$ may provide insights into the time employed by an active droplet to cross a constriction. Indeed, assuming that one simulation timestep corresponds to $T = 10$ ms in real units (further details about the mapping to real values are in Supplementary Note 2), the crossing time $T_c$ ranges approximately between 1.5 h for $\lambda \simeq 0.5$ and 3 h for $\lambda \simeq 0.2$, numbers in qualitative agreement with the ones found, for example, in fibroblasts crossing narrow interstices[44].

These results suggest that, despite the complex physics involved, a single collective variable, measuring the progress of the crossing, is capable of conveying remarkable insights about the process, such as rapidity of the transmigration through different regions of the pore, retraction of the droplet and stationary regimes occurring especially within narrow interstices.

## Discussion

In summary, we have numerically studied the physics of an active gel droplet migrating through a constriction, mimicking conditions potentially reproducible in microfluidic experiments. Key ingredients of the model are the contractility of the polar liquid crystal confined within the drop and the adhesiveness between the fluid interface and the solid surfaces modeling the pore. In addition, hydrodynamic interactions are properly incorporated throughout the model.

We have shown that, if the height $h$ of the constriction is comparable with the diameter $D$ of the droplet, careful control of droplet speed and elasticity (i.e. interfacial tension and elastic deformation of the active fluid) are sufficient to guarantee a smooth crossing. On the contrary, if $h$ becomes considerably smaller than $D$ (i.e. $\lambda \lesssim 0.5$), adhesion forces between the interface of the drop and the pillars of the pore are decisive to enable the crossing. Our results suggest that a stronger adhesion at the entry and a lower one at the exit of the constriction favors the transmigration in conditions that would have been inhibited otherwise. The process entails considerable morphological changes, ranging from crescent to ampule and hourglass-like geometries (structures akin to the ones observed, in similar conditions, in tumor breast cells[59,60]), alongside substantial deformations of the contractile material, including the concurrent presence of splay and bend distortions, the latter generally higher for narrower orifices. The formation of such striking variety of shapes as well as the ordering properties of the contractile material are tightly linked to the fluid–structure, which is found to exhibit two dominating patterns: (i) long-lasting vortices out of the constriction, only temporary surviving within the pore and (ii) short-lived rectilinear flows in the orifice. Hence, the combined effect of confinement and adhesion is to rectify the flow within the constriction, a condition that ultimately enables transmigration. We highlight that the general picture emerging from these results qualitatively holds if a frictional force is included (see Supplementary Note 5). This extra term would mimic, in a phenomenological way, the momentum sink due to the presence of walls placed at an infinitesimally small distance, as in a thin film.

On an experimental side, these results can be reproduced by using microfluidic techniques already adopted to study the transmigration of biological cells[44,61]. Our system could be mapped onto a microchannel of length 0.5–1 mm equipped with PDMS solid pillars, whose surfaces are placed at distances ranging from 10 to 70 μm and functionalized with proteins (such as collagen or fibronectin) favoring droplet adhesion. Two different concentrations of such proteins could model the change of adhesion strength at the entry and exit of the constriction. A self-propelled micrometer droplet of $D \simeq 90\,\mu m$ moving at speed from 1 up to 10 μm/s can be self-assembled by encapsulating an active polar gel (with effective viscosity $\eta_{eff} \simeq 1.5 kPa\,s$ and elastic constant $\kappa \simeq 4\,nN$) within water-in-oil emulsions, following the formulation of ref. [12]. Further details can be found in the "Methods" section and Supplementary Note 3.

Besides providing a deeper understanding of the physics of active fluid droplets migrating in constrained environments, our results may prove useful for the realization of bio-inspired artificial swimmers capable of transporting cargo to specific locations, a process of interest in drug delivery in which one needs to efficiently load pharmaceutical molecules without compromising the structural integrity of the carrier, especially when moving through microscale constrictions. In addition, since some aspects of this active droplet resemble those of laminar cell fragments[54,55], our results could provide insights for ameliorating the design of microfluidic cell sorting devices, which make use of surfaces patterned with specific adhesive properties to detect and isolate cells[56]. Yet, our model remains distant from a living cell in many aspects, such as the lack of the nucleus and of the complex underlying biochemical network governing, for example, the mechanics of the focal adhesions[53]. Such drawback could be partially overcome by considering a slightly more realistic description of a cell, modeled as a double emulsion[62] in which the inner droplet provides a highly simplified representation of a nucleus and the contractile material is confined within the layer, mimicking the tiny cortex of eukaryotic cells containing the actin cytoskeleton. On a biological side, this could be of interest, for example, for studying the effect of physical confinement on tumor cells (such as metastatic breast cancer), where the transmigration has been found to occur even when actin polymerization or myosin contractility is inhibited[57]. In spite of these limitations, our results may support the view that some aspects of droplet migration through constrained environments would strongly rely upon mesoscopic physical ingredients, such as speed, elasticity, and adhesion forces rather than on the microscopic details of the physics involved. We finally mention that an alternative class of model systems, potentially useful for designing artificial swimmers, is that of active vesicles, which are built by encapsulating self-propelled particles within a soft membrane[63–65]. Such objects have been found to reproduce some features of motile cells, including membrane fluctuations and highly branched sub-micrometer protrusions, phenomena occurring at lengthscales usually inaccessible by exclusive mean-field-like approaches but often crucial in driving pathological processes (such as cancer metastasis) within highly confined-environments.

## Methods

### Basic idea of the model

Here we shortly outline the hydrodynamic model used in this work. We consider a self-propelled fluid droplet containing an active polar liquid crystal (or an active gel) immersed in a passive fluid. The active gel concentration is described in terms of a scalar field $\phi_1(\mathbf{r}, t)$, positive within the drop and zero outside. The environment surrounding the droplet is a further passive fluid modeling a wet solvent. Such a mixture is embedded within a microfluidic channel made of two flat parallel walls plus two semi-circular symmetric pillars forming a narrow constriction (see Fig. 1). Unlike the flat walls (implemented using no-slip conditions[66], see Supplementary Note 1), the solid structure of the pillars is modeled using two auxiliary *static* phase fields $\phi_2(\mathbf{r})$ and $\phi_3(\mathbf{r})$, positive within each pillar and zero outside. Although this approach provides an approximate description of a constriction, it allows for the relatively easy computational implementation of mesoscale physical effects occurring between drops and pillars (such as the repulsion and the adhesion of the fluid interface with a wall) by minimally modifying a pore-free model already used in previous studies[20,25]. As mentioned above, our active drop also hosts a contractile gel whose experimental realization is, for example, an acto-myosin solution. Its mesoscale order is captured by a polar liquid crystal field $\mathbf{P}(\mathbf{r}, t)$ representing a coarse-grained average of all orientations of the internal constituents (e.g an actin filament). This vector field is positive within the droplet and zero anywhere else. Finally, a further vector field $\mathbf{v}(\mathbf{r}, t)$ describes the global fluid velocity of both drop and solvent.

## Free energy

The equilibrium properties of a purely passive system are encoded in a coarse-grained free energy density[62]

$$
\begin{aligned}
f = {} & \frac{a}{4\phi_{cr}^4}\sum_i^N \phi_i^2(\phi_i - \phi_0)^2 + \frac{k}{2}\sum_i^N (\nabla\phi_i)^2 \\
& - \frac{\alpha}{2}\sum_i^N \frac{(\phi_i - \phi_{cr})}{\phi_{cr}}|\mathbf{P}_i|^2 + \frac{\alpha}{4}\sum_i^N |\mathbf{P}_i|^4 + \frac{\kappa}{2}\sum_i^N (\nabla\mathbf{P}_i)^2 \\
& + \sum_{i,j,i<j} \epsilon_{ij}\phi_i\phi_j + \sum_{i,j,i<j} \gamma_{ij}\nabla\phi_i\nabla\phi_j,
\end{aligned}
\tag{1}
$$

where $i = 1, 2, 3$ and $N$ is the total number of phases, i.e. the active drop and the two pillars. Note that, since the polarization is confined within the droplet, the sole nonzero term is $\mathbf{P}_1$, whereas $\mathbf{P}_2 = 0$ and $\mathbf{P}_3 = 0$. Hereafter (and in the text as well) we set $\mathbf{P}_1 = \mathbf{P}$.

Equation (1) combines three principal contributions, the first two terms stemming from a typical binary fluid formalism, the following three terms borrowed from liquid crystal theory, and the remaining part gauging the interaction between the active drop and the pore. In particular, the first term of Eq. (1) multiplied by the positive constant $a$ ensures the existence of two coexisting minima, $\phi_i = \phi_{eq}$ inside the $i$th phase and $\phi_i = 0$ outside, while the second one determines the interfacial tension whose strength depends on the positive constant $k$ and reads $\sigma = \sqrt{8ak/9}$. The following contributions, comprising the terms multiplied by the factor $\alpha$, represent the bulk free energy associated with the polar phase expanded up to the fourth order in the polarization $\mathbf{P}$. Here $\phi_{cr} = \phi_0/2$ is the critical concentration at which the transition from the isotropic (everywhere outside the active drop, where $|\mathbf{P}| = 0$) to the polar phase (only within the active drop where $|\mathbf{P}| > 0$) occurs. The term in gradients of $\mathbf{P}$ captures the elastic penalty associated with local distortions of the polar liquid crystal within the standard approximation of the single elastic constant $\kappa$[48]. The penultimate contribution, whose strength is controlled by the coefficients $\epsilon_{ij}$, mimics a repulsive effect essentially penalizing the overlap between the active drop and the pillars while the last term, multiplied by the coefficients $\gamma_{ij}$ and modeling adhesion, favors the contact between them.

In summary, at equilibrium we have the following phases obtained minimizing the free energy $F = \int_V f\, dV$: a passive isotropic fluid (where $\phi_1 = 0$, $\phi_2 = 0$, $\phi_3 = 0$ and $\mathbf{P} = 0$) external to the active drop and to the pillars; a polarized region (where $\mathbf{P} = \mathbf{P}_{eq}$, $\phi_1 = \phi_{eq}$, $\phi_2 = 0$, $\phi_3 = 0$) located solely within the drop containing the contractile material; two solid pillars (where $\phi_1 = 0$, $\mathbf{P} = 0$, with $\phi_2 = \phi_{eq}$ and $\phi_3 = 0$ in the pillar at the top while $\phi_2 = 0$ and $\phi_3 = \phi_{eq}$ in the one at the bottom). The values of $\phi_{eq}$ and $\mathbf{P}_{eq}$ are found by minimizing $F$ in a state of uniform $\phi_i$ and $\mathbf{P}$. Across the interface of the active droplet, the values of $\phi_1$ and $\mathbf{P}$ vary smoothly from $\phi_1 = \phi_0$ and $\mathbf{P} = \mathbf{P}_{eq}$ to $\phi_1 = 0$ and $\mathbf{P} = 0$. Finally, we assume equal repulsion between all phase fields, thus $\epsilon_{ij} = \epsilon$ (with $\epsilon$ fixed at 0.1), and nonzero equal adhesion only between the drop and the two pillars, hence $\gamma_{12} = \gamma_{13}$.

## Equations of motion

On a general basis, the dynamics of the order parameters $\phi_i$ is governed by a set of Cahn–Hilliard equations

$$
\frac{\partial\phi_i}{\partial t} + \nabla\cdot(\phi_i\mathbf{v}) = M\nabla^2\mu_i,
\tag{2}
$$

where $M$ is the mobility and $\mu_i = \delta F/\delta\phi_i$ is the chemical potential. This is the canonical Model B[67] describing the dynamics of a conserved scalar order parameter $\phi$. Note that in our model the evolving phase field is $\phi_1$ which is associated with the active drop, while the other two, $\phi_2$ and $\phi_3$ modeling the pillars, are static. The presence of further active drops

(not considered in this paper) would require the inclusion of an equivalent number of dynamic phase fields.

The evolution equation for the polarization $\mathbf{P}(\mathbf{r}, t)$ is given by[1]

$$
\frac{\partial\mathbf{P}}{\partial t} + (\mathbf{v}\cdot\nabla)\mathbf{P} = -\underline{\underline{\Omega}}\cdot\mathbf{P} + \xi\underline{\underline{D}}\cdot\mathbf{P} - \frac{1}{\Gamma}\frac{\delta F}{\delta\mathbf{P}},
\tag{3}
$$

where $\underline{\underline{D}} = (\underline{\underline{W}} + \underline{\underline{W}}^T)/2$ and $\underline{\underline{\Omega}} = (\underline{\underline{W}} - \underline{\underline{W}}^T)/2$ are the symmetric and antisymmetric parts of the velocity gradient tensor $W_{\alpha\beta} = \partial_\beta v_\alpha$. The constant $\xi$ depends on the geometry of the active particles, it is positive for the rod-like ones and negative for the oblate ones. In addition, it controls the response of such entities under shear, whether they are flow aligning ($|\xi| > 1$) or flow tumbling ($|\xi| < 1$). In the former case (the one considered in this paper), they align along the flow direction at a fixed angle, whereas in the latter they reorient chaotically. As in previous works[20,25], we have set $\xi > 1$. The last term is the molecular field $\mathbf{h} = \delta F/\delta\mathbf{P}$, a quantity governing the relaxation of the liquid crystal towards equilibrium, multiplied by the rotational viscosity $\Gamma$ which sets the time scale of the relaxation.

The fluid velocity $\mathbf{v}$ obeys the Navier–Stokes equations which, in the incompressible limit, are

$$
\nabla\cdot\mathbf{v} = 0,
\tag{4}
$$

$$
\rho\left(\frac{\partial}{\partial t} + \mathbf{v}\cdot\nabla\right)\mathbf{v} = -\nabla p + \nabla\cdot(\underline{\underline{\sigma}}^{active} + \underline{\underline{\sigma}}^{passive}),
\tag{5}
$$

where $\rho$ is the density of the fluid and $p$ is the isotropic pressure. At the right-hand side of Eq. (5), $\underline{\underline{\sigma}}^{active} + \underline{\underline{\sigma}}^{passive}$ is the total stress tensor, given by the sum of two contributions. The first one is

$$
\sigma_{\alpha\beta}^{active} = -\zeta\phi_1\left(P_\alpha P_\beta - \frac{1}{d}|\mathbf{P}|^2\delta_{\alpha\beta}\right),
\tag{6}
$$

where $d$ is the dimension of the system and $\zeta$ is a phenomenological parameter gauging the activity strength, positive for extensile particles and negative for contractile ones[1]. The Greek indexes denote Cartesian components. In our model $\zeta$ is negative, signifying the tendency of the active gel to contract along the direction of the inner units (e.g. the actin filaments). The functional form of Eq. (6) can be derived by summing the contribution of each force dipole (produced using energy coming from, for example, ATP hydrolysis of the myosin) and then coarse graining[4].

Passive stress comprises three contributions, namely a viscous term given by

$$
\sigma_{\alpha\beta}^{viscous} = \eta(\partial_\alpha v_\beta + \partial_\beta v_\alpha)
\tag{7}
$$

where $\eta$ is the shear viscosity, summed with an elastic stress $\underline{\underline{\sigma}}^{elastic}$, due to bulk distortions of the liquid crystal, and a surface tension term $\underline{\underline{\sigma}}^{interface}$. The elastic term is

$$
\sigma_{\alpha\beta}^{elastic} = \frac{1}{2}(P_\alpha h_\beta - P_\beta h_\alpha) - \frac{\xi}{2}(P_\alpha h_\beta + P_\beta h_\alpha) - \kappa\partial_\alpha P_\gamma\partial_\beta P_\gamma,
\tag{8}
$$

while the interfacial one is

$$
\sigma_{\alpha\beta}^{interface} = \sum_i\left[\left(f - \phi_i\frac{\delta\mathcal{F}}{\delta\phi_i}\right)\delta_{\alpha\beta} - \frac{\partial f}{\partial(\partial_\beta\phi_i)}\partial_\alpha\phi_i\right].
\tag{9}
$$

Note that the sum in Eq. (9) is necessary since one has to include the contributions due to the pillars, whereas Eq. (8) solely accounts for those stemming from the liquid crystal confined within the motile droplet.

Equations (2–5) are numerically solved by using a hybrid lattice Boltzmann (LB) approach[66,68], in which a predictor-corrector integration scheme is used for Eqs. (2) and (3) while a standard LB method is employed for Eqs. (4) and (5). This method has been successfully tested for a variety of soft matter systems, ranging from binary fluids[69], liquid crystals[70], and active matter[39,68]. Further details about numerical implementation and thermodynamic parameters can be found in Supplementary Notes 1 and 2.

## Data availability
Necessary information to reproduce the simulated data is provided in the "Methods" section and in Supplementary Information. Data are also available upon request from the authors.

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

## Acknowledgements
A.T., M.D., M.L., A.M., and S.S. acknowledge funding from the European Research Council under the European Union's Horizon 2020 Framework Program (No. FP/2014-2020) ERC Grant Agreement No. 739964 (COP-MAT) and ERC-PoC2 grant No. 101081171 (DropTrack). A.T. and D.M. warmly thank Antonio Basoni, Giuseppe Gonnella, and Alexander Morozov for useful discussions.

## Author contributions
A.T. conceived the project. A.T. and S.S. designed the research with the support of M.D., M.L., and A.M. A.T. run simulations and processed data. A.T. analyzed the results with M.D., M.L., A.M., D.M., and S.S. The paper is written by A.T. with contributions from M.D., M.L., A.M., D.M., and S.S.

## Competing interests
The authors declare no competing interests.
