## [Peer Review File · Nature Communications]

The crucial role of adhesion in the transmigration of active droplets through interstitial orificesREVIEWER COMMENTS

Reviewer #1 (Remarks to the Author):

Referee report on NCOMMS-22-16069: ``The crucial role of adhesion in the transmigration of active droplets through interstitial orifices," by Tiribocchi et al.

The authors perform a series numerical calculations of the motion of an active droplet through a constriction. The active droplet is propelled by self-generated contractile stresses. It was shown previously (references [19] and [24] in the manuscript) that at high enough activity spontaneous symmetry breaking can occur leading to deformation and net motion of an active droplet. The authors use this model for the droplet and couple it to the low-Reynolds number flow in the constriction to see under what conditions -- constriction ratio, activity, level, etc. -- will a droplet be able to migrate across the constriction. They also explore the role of a model of adhesive forces between the droplet surface and the constriction on the transmigration.

The model is well founded and carefully described and the numerical calculations appear to be of high quality. The results are perfectly reasonable -- if the drop is too big and/or the activity too low it can't cross the constriction. If one adds adhesion on one side only this helps the drop get across. What did we learn that was not already apparent at the outset? The spontaneous symmetry breaking was known. Motion of passive drops through constrictions have been studied in the past. Apart from the specific detailed conditions, everything is obvious. So why do the calculations? If there were experimental results that afforded a detailed comparison, then I could see the motivation for the study.

With regard to experiments, a real experiment would be in 3D not 2D, and the role of interfacial tension can differ owing to the second radius of curvature in 3D. If the drop were so large as to fill a microfluidic channel and therefore be (roughly) 2D, then the fluid motion will be governed by Hele-Shaw flow not Stokes flow. So either one needs to do a 3D drop in Stokes flow or a 2D drop in Hele-Shaw flow. Not a 2D drop in Stokes flow.

The work is a solid contribution to the literature, but not of the importance and novelty expected for Nature Communications.

Reviewer #2 (Remarks to the Author):

The authors study the transmigration of droplets that are self-propelled through interstitial orifices. They highlight that motility regimes typical of biological cells are found. Using computer simulations in two dimensions and a phase field-like approach, they predict the motility of droplets containing an active nematic as well as flows that are induced by the activity. Lattice-Boltzmann simulations are used to calculate systems with active droplets in constrained environments. Three phase fields are used. One of the phase fields accounts for the density of active material within the droplet, whereas two static and fluid-free phase fields are used to model pillars that are used to implement a constriction into a straight channel. The authors show that an adhesive interaction between the active droplet and the boundaries of the orifice can enable droplets to pass that have sizes that are larger than the width of the orifice.

The work is original, and on the first view, the results seem relevant for studying biomimetic systems, in particular having in mind biological cells moving through constrictions or--thinking along the lines of synthetic biology--chemically synthesized active droplets that may play a role in pharmacological applications. The methods employed are applied rigorously, and the interaction of the droplets with orifices of various widths is studied systematically. However, I am not entirely sure to which extent the work opens routes that could be followed to realize the predictions experimentally in a synthetic system. Therefore, I think that a more careful discussion on which lessons could be learned for understanding motility in biological systems (and for which conclusions the model may be too simplified) would be especially helpful to allow a broader audience to benefit from the study. Furthermore, a discussion of and comparison of active droplets with active vesicles may help better embed the study into the pertinent literature. Similarly, regarding driving the activity, filament-motor mixtures may be discussed along with motility assays where the filaments are propelled on a bed of motors.

1) The authors mention active gels in the introduction: filament-motor systems, such as microtubule-kinesin and actin-myosin systems [Refs. 10-13]. I wonder whether comparing the results of the active-droplet systems with motile-cell systems, assays where motors propel cytoskeletal filaments on a substrate, such as filaments on a bed of motors, may not be more relevant? Also, but not only because the authors later also refer to bacteria. Studies that come to my mind are:

- Peruani et al., Phys. Rev. E 74, 030904 (2006)
- Schaller et al., Nature 467, 73 (2010)
- Sumino et al., Nature 483, 448 (2012)
- Abkenar et al., Phys. Rev. E 88, 062314 (2013)

2) I think that in addition to the active droplet model, active-vesicle models should be discussed. Like active droplets, they may help understand mechanisms responsible for motility in biological systems. Some publications--mostly theoretical as the manuscript under consideration--are

- Paoluzzi et al., *Sci. Rep.* 6, 1 (2016)
- Abaurrea-Velasco et al., *New J. Phys.* 21, 123024 (2019)
- Fosnarić et al., *Soft Matter* 15, 5319 (2019)
- Vutukuri et al., *Nature* 586, 52 (2020)
- Peterson et al., *Nat. Commun.* 12, 1 (2021)

3) The system with a microchannel and a constriction studied by the authors reminds me of confined cell migration, such as discussed here:

- Brückner et al., *Nat. Phys.* 15, 595 (2019)

4) The authors state that their droplets contain a contractile material and mention actin-myosin systems, see Fig. 1 (d). This may hint at relevance for cell motility. However, in motile cells, the actin filaments are usually also polymerizing. Can the cytoskeleton of motile cells still be considered a contractile material? If not, can the authors provide other relevant experimental systems for that motility has been studied, perhaps based on actin and myosin?

5) On page 3, the authors discuss that the fluid flow surrounding the droplet acquires a quadrupolar structure, preventing net motion. I wonder which role flow and direct propulsion forces with a substrate may play in systems that the authors imagine? In my opinion, specifying the regimes in which the flow-based propulsion discussed in this study (for example, regarding sizes of the droplet and in comparison with direct propulsion by motors on a substrate) are expected to contribute to the motility significantly would be helpful for readers.

6) Publications that come to my mind regarding the role of flows for cell motility--as discussed in the manuscript under consideration--are:

- Stroka et al., *Cell* 157, 611 (2014)
- Li and Sun, *Biophys. J.* 114, 2965 (2018)

Can the predictions of the current model be connected with these studies?

7) I wonder whether the area of the droplets and the length of their circumference change when the droplet passes the constriction. For a finite adhesion energy gain for the contact between the droplet

and the constriction, is the change of the circumference similarly important as the change of the adhesion strength?

8) On page three, the wording "the speed is not high enough to guarantee the crossing" seems problematic because of the small-Re regime. Wouldn't it be more reasonable to talk about a force here?

9) Can the polarisation and the splay shown in Fig. 2 be related to the cytoskeleton structure of motile cells (or cell fragments) squeezing through constrictions? Would this be something that could be testable in experiments?

10) I wonder whether the authors can relate the findings for their 2D systems to cell motility in a 3D polymer matrix? If not, could the method be generalized to study 2D systems in the future?

11) Some of the subfigures in the main text do not have labels; see Figs. 3, 6.

Reviewer #3 (Remarks to the Author):

Review of manuscript:

The crucial role of adhesion in the transmigration of active droplets through interstitial orifices

The paper describes a computational study of a theoretical model to describe the dynamics and migration of active droplets – here modeled as a coarse-grained continuum suspension of polar active agents in an ambient fluid – through channels with and without obstructions. Obstructions are introduced using semi-cylindrical surfaces that constrain crossing droplets to deform and interact with the interface significantly. The surface tension of the composite active droplet and its viscosity are prescribed. The overall state of the interior of the drop is characterized by the density of the agents, their orientations or their polarity, the fluid field (velocity) and the stress that has active and passive components. The active stress is modeled using a phenomenological model based on previous work; here treating the polar agents as exerting stresses that depend on their polarity and their density (concentration). The fluid velocities are modeled using the Navier-Stokes equation for the incompressible system. The passive stress has viscous, elastic and surface components that are modeled using extensions of liquid crystal theory with the interfacial stress invoking surface tension and interfacial energy contributions that include repulsive interactions with possible solid boundaries and possibly attractive adhesive components.

Extensive lattice Boltzmann simulations show a rich variety of dynamic regimes whose physics is controlled by i) constriction geometry, ii) droplet activity and elasticity, and iii) adhesiveness between pillars of the pore and droplet. The authors conclude that adhesion forces are decisive to enable the transmigration, especially for narrow interstices. For wide pores droplets can cross the constriction depending on the balance between droplet speed and elasticity. For narrow pores, a critical adhesion gradient is required to enable the crossing.

The model described in the paper seems theoretically robust, and the simulations show intriguing and novel physics of the active droplets. The results are very relevant and interesting for active matter scientists, people working in microfluidics, soft matter, and fluid mechanics. The fundamental question addressed here is important and has not been studied sufficiently. This paper thus makes an important contribution to the field and is also of general interest to the Nature Communications community.

However, in my view, the paper has gaps and drawbacks that make the current version overtly technical paper as written and not of sufficient generality. Additionally, while specific simulations illustrate the physical mechanisms behind the process, the larger picture as to what this means in terms of motivating experiments or relevance to current experiments is lacking.

The topic is however timely and important. Therefore, I suggest that the authors revise the manuscript and, in the process, address the following questions before resubmission and review.

General questions:

1. The paper uses a prescribed continuum phenomenological model that builds on previous concepts drawn from non-equilibrium thermodynamics of active systems, liquid crystal theory and statistical physics of ordered systems (detailed in the Appendix). This results in several parameters that are poorly described, defined or motivated. Having a list or a table of parameters, their physical significance, and associated dimensionless numbers is critical to aid understanding. Such an approach has worked spectacularly in studying the dynamics of passive drops (with the Weber number, Capillary number, Ohnesorge number, Bond number etc., allowing for a beautiful quantification of the phase plot in terms of bounce/capture/motion). Can the authors come up with a minimal set of dimensional parameters that describes this system theoretically?

2. While the simulations are very illuminating (I particularly like Figures 6 and 7), the discussion is highly descriptive. I would have liked to have seen or have the authors explore possible phase space plots. For instance, when the activity parameters and droplet properties are kept constant and only the

constriction geometry is varying, how does the phase-space plot of non-crossing, crossing and adhesion-enabled look? An analysis of this is important for one to go beyond the parameters just described. Is there a suitable example phase-plot that may help with the “big-picture”?

3. The paper is purely computational. This is of course a new topic and so I understand that generating experimental results will probably be a whole new project. Nonetheless, the authors motivate their study by referencing possible microfluidic applications and inspired by biophysical examples of such self-propelling active droplets. It is important to include a discussion therefore of possible experimental systems that may be used to verify some of the physical mechanisms and features predicted by the computations. What experiments do the authors propose that builds on their results? What would be experimental systems and parameters that map to the parameters studied by the authors? How would one generate an adhesion gradient that serves as a parameter independent of the surface tension of the drops?

4. Model-wise, I was left confused by the fundamental reasoning and motivation underlying the surface tension form used here. In experimental systems of polar particles in fluids such as bacterial swarms and bacterial suspensions, the anchoring of the bacteria at the interface (be this a fluid-fluid interface, or the triple fluid-fluid-solid interface) is as important as the intrinsic activity and density in setting the flow structure inside the droplet. There is barely any mention of this here, and if this is somehow intrinsic to the model, this must be addressed and discussed.

Technical questions:

1. How is this related to micro swimmers? I was also not sure about the geometry of the simulation by just reading the main text. Is the simulation 2D, quasi-2D, 3D? Are the constrictions two dimensional? If it is 2D please motivate the geometry analyzed. If the simulations are in 3D what is the 3D structure of the fluid vortical field in Figure 1 (e,f), and in Figures 5 and 6? If quasi-2D what is the boundary condition imposed in the transverse direction? Typical micro swimmers swim in 3D so clarity is required if extensions to possible micro swimmer applications are suggested.

2. The time scale seemed elusive to me. What is the time (if one were to translate to real units) of the crossing in Figures 6 and 7. Since for very narrow constrictions the droplet may spend a long time within the constriction; this is perhaps a useful quantity to know.

3. Parameters are introduced and assigned numerical values without any explanation or motivation. For instance, why is $\beta = 0.1$? What is the value of β (other than its greater than unity)? What is the rotational viscosity and what's the motivation for the values used? Are these motivated by experiments or do the values correspond to proof-of-concept values?

4. The authors talk of contractility of the droplet. How is this defined? What dimensional number quantifies this and how is this related to the active “softness” or deformability of a droplet translating in unbounded fluid?

5. What exactly is stronger adhesion at entry and lower one at exit mean. Can one vary the adhesion field smoothly in a manner that may enable drops to cross controllably? An example computation that studies this for very narrow channels where the droplet curvatures near the entrance and exit provide length scales that may be compared to the adhesion gradient would be very useful. At least this point should be addressed.

6. Finally, the connection to dipoles generated by the actin-myosin system needs clarification. In the model presented in the figure, the molecular motors generate sliding forces that causes the filaments to slide and contract. What is the embedded medium that generates resistance to the motion of the filaments since motor models typically assume overdamped dynamics? How are the active forces exerted on the ambient medium by these dipole units related to the active stress form assumed in the model? Justification here would help and motivate. Also, many filaments are polar and the way the figure is drawn the filaments will slide until perhaps an equilibrium configuration is reached upon which the motor activity is stalled – is this a correct interpretation? If this is indeed what happens, where does the active component of the stress generated post-stall enter the model? Connecting a conceptual minimal model of possible active polar agents to the computational model and justifying the form of the active stress terms used (with estimates of relevant parameters) will help improve the presentation significantly.

Reply to Referee #1

Comment: *The authors perform a series numerical calculations of the motion of an active droplet through a constriction. The active droplet is propelled by self-generated contractile stresses. It was shown previously (references [19] and [24] in the manuscript) that at high enough activity spontaneous symmetry breaking can occur leading to deformation and net motion of an active droplet. The authors use this model for the droplet and couple it to the low-Reynolds number flow in the constriction to see under what conditions – constriction ratio, activity, level, etc. – will a droplet be able to migrate across the constriction. They also explore the role of a model of adhesive forces between the droplet surface and the constriction on the transmigration.*

The model is well founded and carefully described and the numerical calculations appear to be of high quality. The results are perfectly reasonable – if the drop is too big and/or the activity too low it can't cross the constriction. If one adds adhesion on one side only this helps the drop get across. What did we learn that was not already apparent at the outset? The spontaneous symmetry breaking was known. Motion of passive drops through constrictions have been studied in the past. Apart from the specific detailed conditions, everything is obvious. So why do the calculations? If there were experimental results that afforded a detailed comparison, then I could see the motivation for the study.

Our response: We thank the Reviewer for the positive opinion about the quality of our manuscript and for its critical reading. However, we kindly disagree with some aspects of the criticism which, we believe, oversimplifies the picture emerging from our study.

While it is true that the motion of passive drops through constrictions has been widely studied, the transmigration of active emulsions has never been theoretically investigated as in this work, where i) hydrodynamics is incorporated on a fully consistent basis, ii) the geometry of the channel closely resembles that used in microfluidic experiments of living cells (see, for example, P. M. Davidson et al., "Design of a microfluidic device to quantify dynamic intra-nuclear deformation during cell migration through confining environments" *Integrative Biology* **7**, 1534-1546 (2015)) and iii) the dynamics of the active droplet is designed to mimic the one of experimentally realized self-propelled water-in-oil emulsions (see, for example, T. Sanchez et al., "Spontaneous motion in hierarchically assembled active matter" *Nature* **491**, 431-434 (2013)).

Also, the process is considerably more complex than that described by the Reviewer. If the confinement parameter λ and droplet speed v are sufficiently large, the crossing occurs rather smoothly without the need of adhesion (Fig.2a-f). On the contrary, if λ decreases, a *non-uniform* design of adhesive contacts is fundamental to enable the crossing (Fig2g-v), even for the same droplet speed. In other words, adhesion must have a broken symmetry between inlet and outlet. Importantly, the strength of the adhesive forces has to be carefully balanced, since either an excess or a shortage of adhesion would prevent the crossing. A design like the one proposed in our work could be experimentally realized by coating the pillars with collagen or fibronectin which are known to favour adhesion (see, for example, D. B. Brückner et al., "Stochastic nonlinear dynamics of confined cell migration in two-state systems" *Nat. Phys.* **15**, 595 (2019)). Our simulations also unveil two further important results: i) within the pore, the polar field acquires bend distortions which generally emerge as instabilities associated

to *extensile* materials; ii) adhesion forces and confinement rectify the fluid velocity within the constriction, a result akin to the one observed in microfluidic experiments on motility of tumor cell driven by water permeation (see Stroka et al., "Water permeation drives tumor cell migration in confined microenvironments", *Cell* **157**, 611 (2014)). The reliability of our results is also further supported by other quantitative data, such as transmigration time, droplet speed and elastic parameters, whose values are in agreement with the typical ones of microfluidic experiments of synthetic active drops and cells. These considerations have been highlighted in the revised manuscript.

As a final remark it is worth highlighting that, besides demonstrating the results of existing experiments, the best scope of the simulations is to imagine new routes for the design of the future ones, especially if their feasibility relies upon cross-fertilizing ideas of contiguous fields of research.

Comment: *With regard to experiments, a real experiment would be in 3D not 2D, and the role of interfacial tension can differ owing to the second radius of curvature in 3D. If the drop were so large as to fill a microfluidic channel and therefore be (roughly) 2D, then the fluid motion will be governed by Hele-Shaw flow not Stokes flow. So either one needs to do a 3D drop in Stokes flow or a 2D drop in Hele-Shaw flow. Not a 2D drop in Stokes flow.*

The work is a solid contribution to the literature, but not of the importance and novelty expected for Nature Communications.

Our response: We surely agree with the Reviewer that 3D numerical simulations would be closer to realistic experiments. However, they represent a formidable computational challenge, especially in systems described by multi-dimensional parameter space as in this study. As mentioned in the previous response, the undeniable advantage of 2D simulations is that they allow to capture, at a much reduced computational cost, many features observed in real experiments, such as shape deformations, some aspects of the flow field, transmigration times, droplet speed, to name but a few and, concurrently, to predict behaviors (such as the orientation of the active material) difficult to characterize. We would also like to kindly highlight that a large body of highly cited theoretical works on 2D droplet motility has been published on high impact factor journals over the years, see for example i) Ziebert, F. et al., "Model for self-polarization and motility of keratocyte fragments", *Journ. Roy. Soc. Int.* **9**, 70 (2012); ii) Shao, D. et al., "Coupling actin flow, adhesion, and morphology in a computational cell motility model" *PNAS* **109**, 6851-6856 (2012); iii) Marth, V. and Voigt, A., "Collective migration under hydrodynamic interactions: a computational approach", *Interface Focus* **6**, 20160037 (2016); iv) Camley, B. A. et al., "Polarity mechanisms such as contact inhibition of locomotion regulate persistent rotational motion of mammalian cells on micropatterns", *PNAS* **111**, 14770-14775 (2014); v) Löber et al., "Collisions of deformable cells lead to collective migration", *Scientific Reports* **5**, 9172 (2015).

Finally, our model solves the Navier-Stokes equation which, in the limit of low Reynolds numbers, reduces to the Stokes one. The Hele-Shaw flow occurs in cells where the walls are at infinitesimally small distance. We agree with the Reviewer that thin films are often relevant experimentally. Rather than considering the singular case of Hele-Shaw flow, we have now considered a version of the model where we include a frictional force (of the form

$-b\mathbf{v}$, where b is a constant and \mathbf{v} is the fluid velocity) to account in a phenomenological way for the presence of nearby walls acting as momentum sinks (this is done routinely in active gel theory to study the wet-to-dry transition, in cases where the dry limit is of relevance to experimental studies). The extra term leads to quantitative, but not qualitative results. If $\lambda \simeq 0.5$ and $b < b_c$ with $b_c \simeq 10^{-3}$, our simulations show that the transmigration essentially shares the same features as those described in the main text, whereas for $b > b_c$ the droplet speed diminishes and the crossing is arrested. We have dedicated a paragraph in the supplementary material about these further results. We agree that specifying this is an important addition and thank the Reviewer for prompting us to do so.

Reply to Referee #2

Comment: *The authors study the transmigration of droplets that are self-propelled through interstitial orifices. They highlight that motility regimes typical of biological cells are found. Using computer simulations in two dimensions and a phase field-like approach, they predict the motility of droplets containing an active nematic as well as flows that are induced by the activity. Lattice-Boltzmann simulations are used to calculate systems with active droplets in constrained environments. Three phase fields are used. One of the phase fields accounts for the density of active material within the droplet, whereas two static and fluid-free phase fields are used to model pillars that are used to implement a constriction into a straight channel. The authors show that an adhesive interaction between the active droplet and the boundaries of the orifice can enable droplets to pass that have sizes that are larger than the width of the orifice.*

The work is original, and on the first view, the results seem relevant for studying biomimetic systems, in particular having in mind biological cells moving through constrictions or—thinking along the lines of synthetic biology—chemically synthesized active droplets that may play a role in pharmacological applications. The methods employed are applied rigorously, and the interaction of the droplets with orifices of various widths is studied systematically. However, I am not entirely sure to which extent the work opens routes that could be followed to realize the predictions experimentally in a synthetic system. Therefore, I think that a more careful discussion on which lessons could be learned for understanding motility in biological systems (and for which conclusions the model may be too simplified) would be especially helpful to allow a broader audience to benefit from the study. Furthermore, a discussion of and comparison of active droplets with active vesicles may help better embed the study into the pertinent literature. Similarly, regarding driving the activity, filament-motor mixtures may be discussed along with motility assays where the filaments are propelled on a bed of motors.

Our response: We warmly thank the Reviewer for the careful reading of the paper. Below we provide a point-by-point response to the criticism raised. More specifically, in the revised text we include a discussion about active-vesicle models and describe potential experiments built on our results and inspired by those of motile cells. Also, whenever appropriate, quantitative agreement between our findings and motile cell experiments has been highlighted. Finally, an analysis in terms of dimensionless numbers, sustained by a new simulation campaign, has been proposed.

Comment: *1) The authors mention active gels in the introduction: filament-motor systems, such as microtubule-kinesin and actin-myosin systems [Refs. 10-13]. I wonder whether comparing the results of the active-droplet systems with motile-cell systems, assays where motors propel cytoskeletal filaments on a substrate, such as filaments on a bed of motors, may not be more relevant? Also, but not only because the authors later also refer to bacteria. Studies that come to my mind are:*

- Peruani et al., *Phys. Rev. E* 74, 030904 (2006)
- Schaller et al., *Nature* 467, 73 (2010)
- Sumino et al., *Nature* 483, 448 (2012)

- Abkenar et al., *Phys. Rev. E* 88, 062314 (2013)

Our response: We thank the Reviewer for the suggested references, which are surely relevant for our study. They have been cited in the introduction of the revised manuscript. Concerning the comparison with motile cells, our model would more closely describe a microfluidic experiment involving synthetic active droplets (such as those realized in T. Sanchez et al., "Spontaneous motion in hierarchically assembled active matter" *Nature* **491**, 431-434 (2013)) rather than real cells, although its functioning is clearly inspired by the latter. However, in the revised paper we have commented, whenever appropriate, about connections between our results and features typically observed in the transmigration of living cells. We have also discussed the design of an experimental setup potentially useful for testing our findings.

Comment:2) *I think that in addition to the active droplet model, active-vesicle models should be discussed. Like active droplets, they may help understand mechanisms responsible for motility in biological systems. Some publications—mostly theoretical as the manuscript under consideration—are*

- Paoluzzi et al., *Sci. Rep.* 6, 1 (2016)

- Abaurrea-Velasco et al., *New J. Phys.* 21, 123024 (2019)

- Fosnaric et al., *Soft Matter* 15, 5319 (2019)

- Vutukuri et al., *Nature* 586, 52 (2020)

- Peterson et al., *Nat. Commun.* 12, 1 (2021)

Our response: We thank the Reviewer for the stimulating references provided. We agree that active-vesicle models represent a further promising example of soft material of potential interest for motility in biological systems and for the design of self-motile micro-drops. A short discussion about their use in the context of synthetic active droplets has been included in the final section of the main text.

As a finale remark we note that our model is however different from the ones proposed in the suggested papers, which generally use particle-based simulations (Brownian or Langevin dynamics) to investigate dynamics and shape deformations of active vesicles. In addition, unlike our study, in those papers hydrodynamic interactions and the effect produced by a confined environment are generally neglected.

Comment:3) *The system with a microchannel and a constriction studied by the authors reminds me of confined cell migration, such as discussed here:*

- Brückner et al., *Nat. Phys.* 15, 595 (2019)

Our response: We thank very much the Reviewer for pointing to this excellent manuscript, which has been included in the reference list of the revised paper. In that paper, the authors design a two-state micro-environment consisting of two square adhesive chambers connected by a thin constriction in which a cell performs repeated transitions. Its dynamics under confinement is captured by a Langevin equation in which the noise strength depends on the state of the system.

Concerning the structure of the device, our design is slightly more akin to that proposed in P. M. Davidson et al., "Design of a microfluidic device to quantify dynamic intra-nuclear deformation during cell migration through confining environments" *Integrative Biology* **7**,

1534-1546 (2015) or P. M. Davidson et al., "Nesprin-2 accumulates at the front of the nucleus during confined cell migration", *Embo Reports* **21**, e49910 (2020), where a rather long microchannel comprising PDMS pillars modeling a constriction with adhesive properties (realized using extracellular matrix proteins, such as fibronectin, like in the suggested paper) is considered. In terms of the dynamics of the droplet, we neglect noise effects and, besides the momentum equation, we also have advection-diffusion equations capturing the evolution of orientation (i.e. the polarization) and concentration (i.e. the phase field) of the active material. However, besides such differences, the suggested paper surely describes one of the closest experimental realization to our model and is certainly of great inspiration for future investigations.

Comment:4) *The authors state that their droplets contain a contractile material and mention actin-myosin systems, see Fig. 1 (d). This may hint at relevance for cell motility. However, in motile cells, the actin filaments are usually also polymerizing. Can the cytoskeleton of motile cells still be considered a contractile material? If not, can the authors provide other relevant experimental systems for that motility has been studied, perhaps based on actin and myosin?*

Our response: We thank the Reviewer for raising this point. As correctly pointed out, in motile cells the actin filaments can polymerize, although some experiments show that motility can be solely triggered by myosin contractility, such as in breast tumor cells (see, Poincloux et al., "Contractility of the cell rear drives invasion of breast tumor cells in 3D Matrigel", *PNAS* **108**, 1943-1948 (2011) and E. Tjhung et al., "Spontaneous symmetry breaking in active droplets provides a generic route to motility", *PNAS* **109**, 12381-12386 (2012)). In such situations (thus in the absence of polymerization), the cytoskeleton can be still considered a contractile material since contraction forces are generated by the interaction of myosin with actin filaments. This has been clarified in the revised manuscript.

Comments:5) *On page 3, the authors discuss that the fluid flow surrounding the droplet acquires a quadrupolar structure, preventing net motion. I wonder which role flow and direct propulsion forces with a substrate may play in systems that the authors imagine? In my opinion, specifying the regimes in which the flow-based propulsion discussed in this study (for example, regarding sizes of the droplet and in comparison with direct propulsion by motors on a substrate) are expected to contribute to the motility significantly would be helpful for readers.*

6) *Publications that come to my mind regarding the role of flows for cell motility—as discussed in the manuscript under consideration—are:*

- *Stroka et al., Cell 157, 611 (2014)*
- *Li and Sun, Biophys. J. 114, 2965 (2018)*

Can the predictions of the current model be connected with these studies?

Our response: We thank the Reviewer for these interesting comments. In the section "Fluid structure interactions" we have discussed the typical structure of the flow observed for medium and narrow interstices. While the spontaneous flow out of the pore results solely from the action of the contractility and is characterized by either a stationary symmetric four-vortex structure or a two-vortex propulsive pattern, within the constriction it is affected by

confined environment and adhesive forces. In this region, especially within narrower gaps, we essentially find a dominating pattern characterized by a rectilinear flow plus a single temporary vortex moving towards the rear as the drop squeezes into the pore. Hence, the ultimate effect of the adhesion forces is to rectify the fluid velocity, a condition necessary to enable the transmigration. We have mentioned these aspect in the final section of the revised paper.

The suggested papers study the tumor cell migration generated by water permeation, a mechanism different from the one described in our study where the motion is triggered by contractility. While the paper of Li et al. proposes a mechanical model of the process (including effects of focal adhesion and membrane friction) but it does not incorporate hydrodynamic interactions, the one of Stroka et al. finds that cell displacement is caused by an osmotic pressure difference across the membrane, consisting of a net inflow of water at the leading edge and a net outflow of water at the trailing one (in the absence of actin polymerization). Such a description points to a picture where the flow, as in our case, is unidirectional, although produced by an alternative mechanism. We have commented about these similarities in the paragraph "Fluid structure interactions". We would like to very much thank the Reviewer for suggesting these precious contributions which, we believe, further strengthen our results.

Comment:7) *I wonder whether the area of the droplets and the length of their circumference change when the droplet passes the constriction. For a finite adhesion energy gain for the contact between the droplet and the constriction, is the change of the circumference similarly important as the change of the adhesion strength?*

Our response: This is very interesting point, not fully emphasized in the paper. We very much thank the Reviewer for the question. The area of the droplet (i.e. the phase field ϕ_1) is preserved during the process, since its evolution is governed by a model B-like dynamics, where the order parameter is conserved. On the contrary, the length of its circumference changes, especially when the transmigration occurs through a narrow constriction. This behavior is described by the plots of the elastic free energy F_{el}^{bf} in Fig.6 (top line), in which the increase within the gap essentially indicates a droplet stretching, larger for smaller pores. Yet, although the change of the perimeter of the droplet is necessary for the crossing, it does not guarantee that such an event occurs. In movie M5, for example, we show that, if adhesion forces are uniform at the entry and at the exit of the pore, the drop remains stuck within, despite the circumference considerably augments. On the contrary, a non-uniform adhesion design on both pillars (such as the one shown in Fig.2g-n or movie M4) ensures the crossing, essentially under conditions akin to the previous one (same speed out of the pore, elasticity and gap size). This occurs because adhesion contacts provide the extra kinetic energy necessary to deform the droplet and push it into the constriction, as highlighted in Fig.4 and Fig.5. Here, the magnitude of the velocity field exhibits highest values when the fluid interface connects to the pillars while being considerably lower far from them. These further considerations have been included in the revised version of the main text.

Comment:8) *On page three, the wording "the speed is not high enough to guarantee the crossing" seems problematic because of the small-Re regime. Wouldn't it be more reasonable*

to talk about a force here?

Our response: Yes, we agree with the Reviewer. The sentence has been rephrased in the revised manuscript.

Comment:9) *Can the polarisation and the splay shown in Fig. 2 be related to the cytoskeleton structure of motile cells (or cell fragments) squeezing through constrictions? Would this be something that could be testable in experiments?*

Our response: In this model (as well as in previous ones, such as Tjhung et al., PNAS **109**, 12381-12386 (2012) and Nat. Comm. **6**, 5420 (2015)) the polarization and the associated distortions would capture, at a mesoscale level, the ordering properties of the contractile material, consisting of actin filaments and myosin. Thus the answer is definitely yes, although this model provides only a simplified view of the cytoskeleton (which would also include other elements, such as microtubules and intermediate filaments), since the scope is to describe a highly complex process using a minimal set of physical ingredients typical of cell mechanics. In terms of experiments, the spatial organization of the polarization could be investigated using well-established methods, such as the fluorescence speckle microscopy (see, for example, Wilson, C. A. et al., "Myosin II contributes to cell-scale actin network treadmill through network disassembly", Nature **465**, 373–377 (2010)), in which the direction of the contractile material as well as its movement can be determined by speckle flow tracking as a function of position.

Comment:10) *I wonder whether the authors can relate the findings for their 2D systems to cell motility in a 3D polymer matrix? If not, could the method be generalized to study 2D systems in the future?*

Our response: It is in principle possible to relate some aspects of our 2d findings to cell motility in a 3d environment. For instance, the transmigration times (included in the revised version) are found to increase when the height of the constriction diminishes, a result in agreement with the typical times observed in real 3d cells transmigrating, under controlled conditions, within microfluidic channels. Also, typical shape deformations (such as ampule-like and hourglass geometries) resemble the ones occurring, for example, in tumor breast cells (see H. W. Hou et al., "Deformability study of breast cancer cells using microfluidics", Biomedical Microdevices **11**, 557-564 (2009)), or A. Raj and K. Sen, "Entry and passage behavior of biological cells in a constricted compliant microchannel", RSC Adv. **8**, 20884-20893 (2018)). In addition, our model matches several features of a fully 3d microfluidic experiments: for example, the typical size of constriction (designed to mimic the non-continuous spatial constraints of physiological microenvironments), the one of the droplet, plus its speed and elasticity, are close to values obtained from experiments. In particular, the speed decrease within the pore and the following increase at the exit reproduce qualitatively well the peak of the velocity of drops/cells leaving constrictions. These further considerations have been included in the revised text.

Our model can be surely extended to study 3d systems, and this is actually our plan of future investigation. For cell motility in a 3D polymeric matrix, one should probably consider a fluid with three different viscosities, matching those of the nucleus, the cell membrane and the extracellular matrix. This will likely affect the structure of the velocity field and potentially

the shape deformations of the cell. An even more exciting route would be the design of 3d multiscale simulations capable of capturing the physics at lower lengthscales, such as the dynamics of filopodia grabbing the polymer filaments of the extracellular matrix to enable cell motion.

Comment: 11) *Some of the subfigures in the main text do not have labels; see Figs. 3, 6.*

Our response: Thank you for spotting them. Labels are included in the revised version.

Reply to Referee #3

Comment: *The paper describes a computational study of a theoretical model to describe the dynamics and migration of active droplets – here modeled as a coarse-grained continuum suspension of polar active agents in an ambient fluid – through channels with and without obstructions. Obstructions are introduced using semi-cylindrical surfaces that constrain crossing droplets to deform and interact with the interface significantly. The surface tension of the composite active droplet and its viscosity are prescribed. The overall state of the interior of the drop is characterized by the density of the agents, their orientations or their polarity, the fluid field (velocity) and the stress that has active and passive components. The active stress is modeled using a phenomenological model based on previous work; here treating the polar agents as exerting stresses that depend on their polarity and their density (concentration). The fluid velocities are modeled using the Navier-Stokes equation for the incompressible system. The passive stress has viscous, elastic and surface components that are modeled using extensions of liquid crystal theory with the interfacial stress invoking surface tension and interfacial energy contributions that include repulsive interactions with possible solid boundaries and possibly attractive adhesive components.*

Extensive lattice Boltzmann simulations show a rich variety of dynamic regimes whose physics is controlled by i) constriction geometry, ii) droplet activity and elasticity, and iii) adhesiveness between pillars of the pore and droplet. The authors conclude that adhesion forces are decisive to enable the transmigration, especially for narrow interstices. For wide pores droplets can cross the constriction depending on the balance between droplet speed and elasticity. For narrow pores, a critical adhesion gradient is required to enable the crossing.

The model described in the paper seems theoretically robust, and the simulations show intriguing and novel physics of the active droplets. The results are very relevant and interesting for active matter scientists, people working in microfluidics, soft matter, and fluid mechanics. The fundamental question addressed here is important and has not been studied sufficiently. This paper thus makes an important contribution to the field and is also of general interest to the Nature Communications community.

However, in my view, the paper has gaps and drawbacks that make the current version overtly technical paper as written and not of sufficient generality. Additionally, while specific simulations illustrate the physical mechanisms behind the process, the larger picture as to what this means in terms of motivating experiments or relevance to current experiments is lacking.

The topic is however timely and important. Therefore, I suggest that the authors revise the manuscript and, in the process, address the following questions before resubmission and review.

Our response: We warmly thank the Reviewer for the careful reading of the manuscript and for deeming it worth of being considered for Nature Communications. In the revised manuscript, we have included a discussion of the physics in terms of suitable dimensionless numbers and run a new simulation campaign aimed at providing a more general picture of the transmigration through medium size and narrow constrictions. Further simulations

have been also dedicated to discussing the transmigration with a smoother adhesion at the pillars. Finally, we have provided further details about connections between our simulations and microfluidic experiments, also highlighting, whenever possible, a quantitative agreement.

Comment: *General questions:*

1. *The paper uses a prescribed continuum phenomenological model that builds on previous concepts drawn from non-equilibrium thermodynamics of active systems, liquid crystal theory and statistical physics of ordered systems (detailed in the Appendix). This results in several parameters that are poorly described, defined or motivated. Having a list or a table of parameters, their physical significance, and associated dimensionless numbers is critical to aid understanding. Such an approach has worked spectacularly in studying the dynamics of passive drops (with the Weber number, Capillary number, Ohnesorge number, Bond number etc., allowing for a beautiful quantification of the phase plot in terms of bounce/capture/motion). Can the authors come up with a minimal set of dimensional parameters that describes this system theoretically?*

Our response: We thank the Reviewer for pointing this out. In the revised version of the section Results, we have included a brand new paragraph where the dynamics of the active droplet is described in terms of selected dimensionless quantities. Besides Reynolds and capillary numbers (which have been kept fixed), we consider the following ones: the Ericksen number, defined as $Er = \zeta R^2 / \kappa$ which quantifies the balance between activity and elasticity of the liquid crystal, the ratio between adhesion coefficients $A = \gamma_L / \gamma_R$ and the ratio between inertial and adhesive forces $I_{A,L,R} = \rho v^2 R^2 / \gamma_{L,R}$. An analysis in terms of the latter shows that, at the entry, inertial forces are much weaker than adhesive ones, since these ones should be high enough to keep the droplet attached to the pillars to enable the crossing. On the contrary, at the exit lower adhesion forces allow a droplet with sufficiently high speed to escape from the pore. Further simulations show that for $\lambda \simeq 0.5$ one has $2 \lesssim A \lesssim 10$ while for $\lambda \simeq 0.2$ one gets $2 \lesssim A \lesssim 3$. Also, for $\lambda \simeq 0.5$ and $\gamma_R = 5 \times 10^{-3}$, one has $I_{A,R} \simeq 0.25$ and $0.025 \lesssim I_{A,L} \lesssim 0.06$, whereas a shorter interval of values is found if $\lambda \simeq 0.2$.

Comment: 2. *While the simulations are very illuminating (I particularly like Figures 6 and 7), the discussion is highly descriptive. I would have liked to have seen or have the authors explore possible phase space plots. For instance, when the activity parameters and droplet properties are kept constant and only the constriction geometry is varying, how does the phase-space plot of non-crossing, crossing and adhesion-enabled look? An analysis of this is important for one to go beyond the parameters just described. Is there a suitable example phase-plot that may help with the “big-picture”?*

Our response: We thank the Reviewer for the question. A systematic study of the phase diagram in this multi-dimensional parameter space is highly demanding computationally, and hence outside the scope of the current work. However, as suggested by the Reviewer, a more general picture can be obtained by varying a restricted number of parameters. We incidentally note that, changing the size of the constriction (i.e. λ) as suggested, entails a concurrent modification of the adhesion strength at the entry and exit of the pore (i.e.

γ_L and γ_R). In other words, the adhesion strength should be tuned once the size of the constriction is changed. However, except this caveat, further simulations have shown, for example, that, if $\gamma_R = 0.01$ and $0.02 < \gamma_L < 0.03$, the crossing occurs through both medium ($\lambda \simeq 0.5$) and narrow ($\lambda \simeq 0.2$) interstices. Such considerations have been included in the revised text and discussed in terms of the aforementioned dimensionless numbers.

Comment:3. *The paper is purely computational. This is of course a new topic and so I understand that generating experimental results will probably be a whole new project. Nonetheless, the authors motivate their study by referencing possible microfluidic applications and inspired by biophysical examples of such self-propelling active droplets. It is important to include a discussion therefore of possible experimental systems that may be used to verify some of the physical mechanisms and features predicted by the computations. What experiments do the authors propose that builds on their results? What would be experimental systems and parameters that map to the parameters studied by the authors? How would one generate an adhesion gradient that serves as a parameter independent of the surface tension of the drops?*

Our response: Potential experiments inspired by our simulations may be realized partially following the design adopted to study the transmigration of biological cells (see, for example, P. M. Davidson et al., "Design of a microfluidic device to quantify dynamic intra-nuclear deformation during cell migration through confining environments" *Integrative Biology* **7**, 1534-1546 (2015) and P. M. Davidson et al., "Nesprin-2 accumulates at the front of the nucleus during confined cell migration" *Embo Reports* **21**, e49910 (2020)).

The device would consist of a microfluidic channel with PDMS solid pillars functionalized with adhesive proteins (such as collagen or fibronectin) to favour droplet adhesion. Two different concentration of such proteins could model the change of adhesion strength at the entry and exit of the constriction. A self-propelled micrometer droplet may be self assembled by encapsulating an active polar gel within a water-in-oil emulsions, following the formulation of T. Sanchez et al., "Spontaneous motion in hierarchically assembled active matter", *Nature* **491**, 431-434 (2012).

A mapping with real units has been described in Supplementary Note 2. Our simulation would approximately correspond to a microfluidic channel of length 0.5-1mm in which the size of the constriction ranges between $\simeq 10\mu\text{m}$ to $\simeq 70\mu\text{m}$. A droplet of diameter $D \simeq 90\mu\text{m}$ and surface tension $\sigma \simeq 4\text{mN/m}$ would roughly move with speed $1 \div 10 \mu\text{m/s}$ in a Newtonian fluid, such as water. The active gel would have an effective shear viscosity $\eta_{eff} \simeq 1.5\text{kPa}\cdot\text{s}$, an effective elastic constant $\kappa \simeq 0.04\text{nN}$ and a rotational viscosity $\Gamma \simeq 1\text{kPa}\cdot\text{s}$. Such considerations have been included in the closing section of the revised manuscript.

Comment:4. *Model-wise, I was left confused by the fundamental reasoning and motivation underlying the surface tension form used here. In experimental systems of polar particles in fluids such as bacterial swarms and bacterial suspensions, the anchoring of the bacteria at the interface (be this a fluid-fluid interface, or the triple fluid-fluid-solid interface) is as important as the intrinsic activity and density in setting the flow structure inside the droplet. There is barely any mention of this here, and if this is somehow intrinsic to the model, this must be addressed and discussed.*

Our response: This is a very good point. As correctly noted by the Reviewer, in our simu-

lations we do not set any specific anchoring condition (whether perpendicular or tangential) at the fluid interface. More specifically, there is no explicit term in the free energy that fixes it. Nonetheless, a preferential orientation at the interface is induced by the active stress, which is generally known to favour a perpendicular anchoring if the mixture is contractile and a tangential one if extensile (see, for instance, M. L. Blow and J. M. Yeomans, "Biphasic, Lyotropic, Active Nematics" Phys. Rev. Lett. **113**, 248303 (2014)). The mechanism is essentially the following. For contractile activity, the droplet stretches perpendicularly to the polarization (initially set parallel to the y -direction, see Fig.1a,b) which, in turn, orients perpendicularly to the interface, except at the ends of the elongated drop (i.e. at its top and bottom). Such orientation persists once the droplet acquires spontaneous motion, a regime where splay deformations dominate. However, the direction of the polar field changes as the drop crosses the constriction, especially within very narrow ones where surprisingly bend deformations prevail. In the result section of the revised paper, we have extended the discussion about such an "induced" anchoring, also highlighting the effect produced, on its orientation, by a narrow constriction.

Comment: *Technical questions:*

1. How is this related to micro swimmers? I was also not sure about the geometry of the simulation by just reading the main text. Is the simulation 2D, quasi-2D, 3D? Are the constrictions two dimensional? If it is 2D please motivate the geometry analyzed. If the simulations are in 3D what is the 3D structure of the fluid vortical field in Figure 1 (e,f), and in Figures 5 and 6? If quasi-2D what is the boundary condition imposed in the transverse direction? Typical micro swimmers swim in 3D so clarity is required if extensions to possible micro swimmer applications are suggested.

Our response: Thank you for the questions. We consider a two dimensional microchannel made of flat walls plus two pillars, modeled as static phase fields. Hence, the geometry and the constriction are 2D. Also, the droplet is modeled as a two dimensional circular region containing a polar active gel confined by interfacial tension. As previously mentioned, the design of the microchannel closely mimics experimentally realized devices, such as the one in P. M. Davidson et al., Integrative Biology **7**, 1534-1546 (2015). Such details are described in Supplementary Note 1 and 2 and have been included in the main text. We finally note that, while we agree that micro-swimmers swim in 3d, many relevant theoretical studies concerning the modeling of bio-inspired micro-droplets have been formulated in 2D, such as

- Ziebert, F. et al., "Model for self-polarization and motility of keratocyte fragments", Journ. Roy. Soc. Int. **9**, 70 (2012);
- Shao, D. et al., "Coupling actin flow, adhesion, and morphology in a computational cell motility model" PNAS **109**, 6851-6856 (2012);
- Marth, V. and Voigt, A., "Collective migration under hydrodynamic interactions: a computational approach", Interface Focus **6**, 20160037 (2016);
- Camley, B. A. et al., "Polarity mechanisms such as contact inhibition of locomotion regulate persistent rotational motion of mammalian cells on micropatterns", PNAS

111, 14770-14775 (2014);

- G. de Magistris et al., "Spontaneous motility of passive emulsion droplets in polar active gels", *Soft Matter* **10**, 7826-7837 (2014).

Comment:2. *The time scale seemed elusive to me. What is the time (if one were to translate to real units) of the crossing in Figures 6 and 7. Since for very narrow constrictions the droplet may spend a long time within the constriction; this is perhaps a useful quantity to know.*

Our response: Thank you for pointing this out. The crossing times can be obtained following the mapping between real units and simulation ones described in Supplementary Note 2. Assuming that one simulation time-step corresponds to $T = 10\text{ms}$ in real units, one can estimate, for example, that the crossing time T_c is $\simeq 7 \times 10^3\text{s}$ (i.e. ~ 2 hours) for $\lambda \simeq 0.3$ and $\simeq 10^4\text{s}$ (i.e. ~ 3 hours) for $\lambda \simeq 0.2$. This is in qualitative agreements with transmigration times observed in real cells. In P. M. Davidson et al., *Integrative Biology* **7**, 1534-1546 (2015), for instance, a fibroblast of diameter $\sim 100\mu\text{m}$ and containing a cell nucleus (usually a limiting factor) crosses a constriction of size $\sim 10\mu\text{m}$ in approximately 6 – 7 hours (see Fig.3 of that paper). Such considerations have been included in the revised manuscript.

Comment:3. *Parameters are introduced and assigned numerical values without any explanation or motivation. For instance, why is $\epsilon = 0.1$? What is the value of ξ (other than its greater than unity)? What is the rotational viscosity and what's the motivation for the values used? Are these motivated by experiments or do the values correspond to proof-of-concept values?*

Our response: Parameters have been selected in order to combine good numerical stability and realistic mapping to physical units (typical of a microfluidic experiment). In particular, the constant ϵ is set to 0.1, a value large enough to prevent the merging of different phase fields. The parameter ξ controls the response of polar particles under flow, and it is larger than 1 for flow aligning particles. In our simulations we have set $\xi = 1.1$ (this is reported in Supplementary Note 1). Finally, the rotational viscosity sets the time scale for the relaxation of the polarization and describes the viscous torque associated with its rotation. Its value is kept fixed to 1. In Supplementary Note 1 we have commented more about their choice, while in Supplementary 2 note we have provided further details about their mapping.

Comment:4. *The authors talk of contractility of the droplet. How is this defined? What dimensional number quantifies this and how is this related to the active "softness" or deformability of a droplet translating in unbounded fluid?*

Our response: The contractility is modeled through the active stress $\sigma_{\alpha\beta}^{\text{active}} = -\zeta\phi_1(P_\alpha P_\beta - 1/d|\mathbf{P}|^2\delta_{\alpha\beta})$ (see Eq.6 of the main text), where the activity ζ sets the strength of this stress. Negative values of ζ describe contractile materials, while positive values account for extensile ones. A dimensionless number quantifying the balance between activity and elasticity of the liquid crystal is the Ericksen number, $Er = \zeta R^2/\kappa$. If Er is sufficiently large (such as larger than 5), contractility overcomes resistance to deformation mediated by the elastic constant

κ . Under these conditions, splay deformations emerge and the droplet is set into motion driven by a double-vortex spontaneous flow. A detailed investigation about the role played by parameter in droplet locomotion has been discussed in E. Tjhung et al., PNAS **109** 12381-12386, (2012). In our simulations, we have $|\zeta|$ ranging from 10^{-4} to 10^{-3} , $R = 45$ and $\kappa = 4 \times 10^{-2}$, thus Er goes from $\simeq 5$ to $\simeq 50$. As mentioned in a previous comment, a discussion in terms of this number and other dimensionless quantities has been included in a new paragraph in the section Results.

Comment:5. *What exactly is stronger adhesion at entry and lower one at exit mean. Can one vary the adhesion field smoothly in a manner that may enable drops to cross controllably? An example computation that studies this for very narrow channels where the droplet curvatures near the entrance and exit provide length scales that may be compared to the adhesion gradient would be very useful. At least this point should be addressed.*

Our response: We thank the Reviewer for the observation. We patterned the pillars with two different adhesion coefficients, a higher one at the entry (covering half of the pillars, see magenta curved line in Fig.2g) and a lower one at the exit (covering the remaining part, see light blue line in Fig.2g). Thus, the droplet would be subject to a larger attraction at the entry, necessary to squeeze within the pore, and a smaller one at the exit, crucial to favour its detachment and the following leave from the constriction. Such a minimal design could, in principle, be easy to realize in microfluidic experiments using adhesive proteins (see, for example, D. B. Brückner et al., "Stochastic nonlinear dynamics of confined cell migration in two-state systems" Nat. Phys. **15**, 595 (2019), where the constriction is patterned with fibronectin). A smoother pattern could be modeled using a position-dependent adhesion coefficient, whose functional form is $\gamma(y) = 0.5[(\gamma_L + \gamma_R) + (\gamma_L - \gamma_R) \tanh((-y + y_0)/a)]$, where y_0 and a are two constants controlling position and width of the slope around the midline of the constriction. In our simulations, a is varied from 1 (steepest slope, approaching a step function) to 20 (mildest slope). Our results show, for example, that, for $\lambda \simeq 0.5$ and $a \lesssim 15$, the transmigration occurs in a way akin to that described in the paper, although the crossing time considerably increases. On the contrary, for higher values of a , the drop gets trapped within the pore. Similar results are observed for $\lambda \simeq 0.2$. These considerations have been added in a separate paragraph in the supplementary material.

Comment:6. *Finally, the connection to dipoles generated by the actin-myosin system needs clarification. In the model presented in the figure, the molecular motors generate sliding forces that causes the filaments to slide and contract. What is the embedded medium that generates resistance to the motion of the filaments since motor models typically assume overdamped dynamics? How are the active forces exerted on the ambient medium by these dipole units related to the active stress form assumed in the model? Justification here would help and motivate. Also, many filaments are polar and the way the figure is drawn the filaments will slide until perhaps an equilibrium configuration is reached upon which the motor activity is stalled – is this a correct interpretation? If this is indeed what happens, where does the active component of the stress generated post-stall enter the model? Connecting a conceptual minimal model of possible active polar agents to the computational model and justifying the form of the active stress terms used (with estimates of relevant parameters) will help improve*

the presentation significantly.

Our response: We thank the Reviewer for the questions raised. As correctly pointed out, the molecular motors generate sliding forces that cause the actin filaments to contract. In our model, such actomyosin suspension (i.e. the active material) is confined within the droplet and is embedded in a Newtonian fluid, whose viscosity is equal to that of the fluid surrounding the droplet. The higher viscosity of the active gel is basically captured by the rotational viscosity Γ , appearing in polar field equation. The active forces are exerted along opposite directions and essentially in parallel to the actin filaments (this is highlighted by the two tick arrows in Fig.1d). This produces the four-vortex structure of the fluid velocity (highlighted by the rounded thin arrows in Fig.1d), which pulls the medium equatorially and expels it axially.

Also, the interpretation about how the active stress functions is the following. In a perfectly ordered, unbounded, parallel collection of contractile particles, the active forces would cancel each other. On the contrary, in a bounded domain (such as that of a droplet) this perfect balance of forces is lost and a droplet containing a contractile fluid oriented as in Fig.1a would initially stretch, temporarily stalling in a configuration like the one of Fig.1b. Such equilibrium state results from a balance between the active stress, which favours the rise of spontaneous flows destabilizing the droplet, and elasticity of the droplet (due to droplet interface and liquid crystal), which oppose shape deformations. If the active parameter ζ is sufficiently high, the active stress overcomes the resistance to deformations, and the spontaneous flow becomes strong enough to induce splay distortions and droplet motion. Importantly, there is no "active component of the stress generated post-stall"; the active stress is fixed from scratch within the droplet at values such that the splay instability can occur. For lower values of ζ , such that $Er = \zeta R^2/\kappa \leq 1$, the droplet would slightly elongate and stall indefinitely in that configuration. These further details have been included in section Results (Fig.1 has been modified) and Methods of the revised text.

Finally, the functional form of the active stress of Eq.6 of the main text is obtained by summing the force dipoles due to each active particle and then coarse graining. The details of these calculations can be found in S. Hatwalne et al., "Rheology of Active-Particle Suspensions", Phys. Rev. Lett. **92**, 118101 (2004).

REVIEWERS' COMMENTS

Reviewer #2 (Remarks to the Author):

The authors have adequately dealt with my issues. I recommend the publication in Nature communications.

Reply to Referee #2

Comment: *The authors have adequately dealt with my issues. I recommend the publication in Nature communications.*

Our response: We warmly thank the Referee for recommending the publication of our manuscript in Nature Communications.